# Rapid electron transfer via dynamic coordinative interaction boosts quantum efficiency for photocatalytic $CO_2$ reduction

Jia-Wei Wang [1], Long Jiang [1], Hai-Hua Huang [1], Zhiji Han [1] & Gangfeng Ouyang [1,2,3✉]

The fulfillment of a high quantum efficiency for photocatalytic $CO_2$ reduction presents a key challenge, which can be overcome by developing strategies for dynamic attachment between photosensitizer and catalyst. In this context, we exploit the use of coordinate bond to connect a pyridine-appended iridium photosensitizer and molecular catalysts for $CO_2$ reduction, which is systematically demonstrated by [1]H nuclear magnetic resonance titration, theoretical calculations, and spectroscopic measurements. The mechanistic investigations reveal that the coordinative interaction between the photosensitizer and an unmodified cobalt phthalocyanine significantly accelerates the electron transfer and thus realizes a remarkable quantum efficiency of 10.2% ± 0.5% at 450 nm for photocatalytic $CO_2$-to-CO conversion with a turnover number of 391 ± 7 and nearly complete selectivity, over 4 times higher than a comparative system with no additional interaction (2.4%±0.2%). Moreover, the decoration of electron-donating amino groups on cobalt phthalocyanine can optimize the quantum efficiency up to 27.9% ± 0.8% at 425 nm, which is more attributable to the enhanced coordinative interaction rather than the intrinsic activity. The control experiments demonstrate that the dynamic feature of coordinative interaction is important to prevent the coordination occupancy of labile sites, also enabling the wide applicability on diverse non-noble-metal catalysts.

[1] KLGHEI of Environment and Energy Chemistry, School of Chemistry, Sun Yat-sen University, Guangzhou, China. [2] Chemistry College, Center of Advanced Analysis and Gene Sequencing, Zhengzhou University, Zhengzhou, China. [3] Guangdong Provincial Key Laboratory of Emergency Test for Dangerous Chemicals, Guangdong Institute of Analysis (China National Analytical Center Guangzhou), Guangzhou, China. ✉email: cesoygf@mail.sysu.edu.cn

Harvesting sunlight for the reductive transformation of $CO_2$ into carbonaceous fuels has been regarded as a sustainable pathway to manufacture carbon-neutral energy, as well as the promising mitigation of greenhouse gas. With the advantages of well-defined and highly modulable molecular structures, many chemists have devoted themselves to the design of molecular systems of light-stimulated $CO_2$ reduction with transition metal complexes, which can function as either photosensitizer (PSs) to transporting electrons upon light excitation, or catalysts to accept electrons for catalytic $CO_2$ reduction[1–8]. Extensive efforts have been dedicated to achieving high catalytic rates[9–13], i.e., turn-over frequency (TOF) or turn-over number (TON), whereas quantum efficiency (QE) also deserves significant attention as it has been considered as the prominent, intrinsic parameter to evaluate the photon-to-product efficiency for a photocatalytic system[14]. The accomplishment of a high QE is challenging as it is limited by multiple factors in a photocatalytic system, including the catalytic performance of the catalyst, the photophysical properties of PS, and also importantly, the electron transfer between PS and catalyst[15]. As a representative example to attain considerable QEs, Kojima et al.[16] designed a Ni catalyst featuring a bio-mimic $S_2N_2$-type tetradentate ligand to couple with $[Ru(bpy)_3]^{2+}$ (bpy = 2,2′-bipyridine) PS, achieving TONs > 700 and a good QE of 1.42% at 450 nm. Afterward, they further decorated the above Ni catalyst with pyridine arms, obtaining $Ni(beptpy_2)^{2+}$ (bpetpy$_2$ = bis(2-pyridyl-3-pyridylmethyl)-1,2-ethanedithiol), which can stabilize an $Mg^{2+}$ ion as a Lewis acid to enhance the fixation of $CO_2$ for further catalysis, reaching a much higher QE of 11.1% at 450 nm for $CO_2$ reduction to CO[17], a record high value to our knowledge among the molecular systems for photocatalytic $CO_2$-to-CO conversion employing noble-metal PSs and earth-abundant catalysts (Supplementary Table 1). Besides the labors in catalyst design, the optimization of PSs also attracts much attention, which can be exemplified by a ring-shaped Re trinuclear complex prepared from Ishitani and co-workers[18]. This Re compound can be utilized as a potent PS to promote the catalytic activity of a Re catalyst, $fac$-$[Re(bpy)(CO)_3(CH_3CN)]^+$, ultimately accomplishing a high QE of 82% of CO formation from $CO_2$ photoreduction at 436 nm. Consequently, the above documents demonstrate that major progress has been made in improving the QEs of molecular systems for $CO_2$ reduction by rational design of catalysts or PSs, based on which, presumably, the QE can be further optimized by strengthening the electron transfer between PS and catalyst.

In this context, effective strategies to accelerate the electron transfer between PS and catalyst are highly desirable. Recently, the construction of supramolecular photocatalytic systems by connecting PS and catalyst via covalent bonds has been exploited (Fig. 1)[19–22]. Pioneering instances include the covalent attachments between Re-/Ru-bpy catalysts and diverse PS moieties by Ishitani et al., generally showing rapid intramolecular charge transfer and accomplishing high QEs[19–21,23,24]. In addition, a very recent example reports the synthesis of a Zn-porphyrin PS dangling with a Re-phen catalytic module, which displays TONs >1300 and a QE of 8% at 420 nm for highly selective $CO_2$-to-CO photo-conversion[24]. However, substantial synthetic efforts are required to prepare such dyad molecules. Moreover, the possibly vulnerable covalent linkers (like acetylene or amide bonds)[25,26] and back electron transfer via covalent connection[27–29] may lead to irreversible losses of stability and activity.

Compared with traditional covalent strategies, a dynamic binding between PS and catalyst may offer an elastic endurance and ultimately a stable performance, as the reversible character of dynamic interaction may enable a self-correction behavior[30–34]. Nevertheless, the documented dynamic assembly for photo-catalytic $CO_2$ reduction is rare, except for an H-bond-interacted system from Kubiak et al.[35] in which a remarkable QE of 23.3%

±0.8% and a TON of 100 ± 4 were achieved (Fig. 1). In addition, the detailed characterizations on the dynamic interaction and mechanistic studies on the facilitated intermolecular electron transfer are elusive, which is worthy of in-depth investigations to provide insights for further optimization on such systems. Consequently, the exploration of strategies to bridge dynamic interaction between PS and catalyst is in urgent need.

As a type of covalent bond, the coordinate bond can be an ideal intermolecular force to attach PS and catalyst. The labile feature will endow the dynamic stability in the coordinatively interacted system. Meanwhile, besides the conventional outer-sphere electron transfer by the random collision between PS and catalyst in a mixture, the intermolecular electron transfer can be additionally facilitated in a manner of inner-sphere electron transfer (ISET) via the coordinative interaction[36] as either a stable adduct or a transition state, potentially overcoming the diffusion limit[37]. Such coordinative interaction between PS and catalyst has been utilized in the molecular systems for hydrogen evolution[31,33,38–40] or immobilizing molecular catalysts on heterogeneous materials[41–44]. The coordinative interaction between tertiary amine as potential sacrificial reagent and planar PSs is also documented[28,40]. To our knowledge, however, it is rare to employ the coordinative interaction between PS and catalyst in a fully molecular photocatalytic system for $CO_2$ reduction.

In this work, we have designed a combination between a pyridine-pendented Ir(III) PS, $[Ir(ppy)_2(qpy)](BF_4)$ (IrQPY; ppy = 2-phenylpyridine; qpy = 4,4′:2′,2″:4″,4‴-quaterpyridine), and a pristine cobalt phthalocyanine (CoPc) catalyst (Fig. 1). The pyridine pendants of IrQPY can serve as the binding sites to axially coordinate to CoPc, a molecular catalyst seldom utilized in photocatalytic $CO_2$ reduction but extensively employed in electrocatalysis[45–49]. As expected, the dynamic coordinative interaction between IrQPY and CoPc has achieved an impressive performance in photocatalytic $CO_2$ reduction, affording a QE of 10.2% ± 0.5% at 450 nm for CO production, with a TON of 391 ± 7 and a high selectivity of 98%, far superior to the IrBPY/CoPc (IrBPY = $[Ir(ppy)_2(bpy)](BF_4)$) system which has no coordinative interaction. Moreover, the coordinative interaction strategy is highly applicable to other molecular catalysts with varying configurations. The selection of an animo-appended CoPc catalyst has further achieved an exceptionally high QE of 27.9% ± 0.8% at 425 nm, where the stronger coordinative binding essentially accounts for the enhanced activity. The above advantages strongly recommend the coordinative interaction as a versatile strategy to improve photocatalytic performance.

## Results

**$^1$H NMR titration for coordinative interaction**. Here, we chose an Ir(III) PS, IrBPY, as the potential prototype. This choice is based on the stronger reducing forces and higher stability of the Ir-based PSs, like IrBPY or $Ir(ppy)_3$, in contrast to the Ru-based PSs like $[Ru(bpy)_3]^{2+}$ or $[Ru(phen)_3]^{2+}$[6]. The decoration of IrBPY with pyridine pendants rendered the structure of IrQPY. The synthetic methods of IrQPY and IrBPY are shown in the Supplementary Information, and CoPc is commercially available. First, we investigated the interaction between IrQPY and CoPc by $^1$H NMR titration in DMF-$d_7$ ((Fig. 2a). The $^1$H NMR spectrum of IrQPY (Supplementary Fig. 1) reveals a local $C2$ symmetry around the Ir(III) center, consistent with the structure obtained from single-crystal X-ray diffraction (Supplementary Fig. 2). The proton positions were assigned with the assistance of $^1$H-$^1$H 2D correlation spectroscopy method (Supplementary Fig. 3). The spectrum of CoPc shows no apparent signal <20 ppm due to the paramagnetic Co$^{II}$. Notably, the titration of CoPc into IrQPY induced slightly shifted, broadened peaks, and faded peak

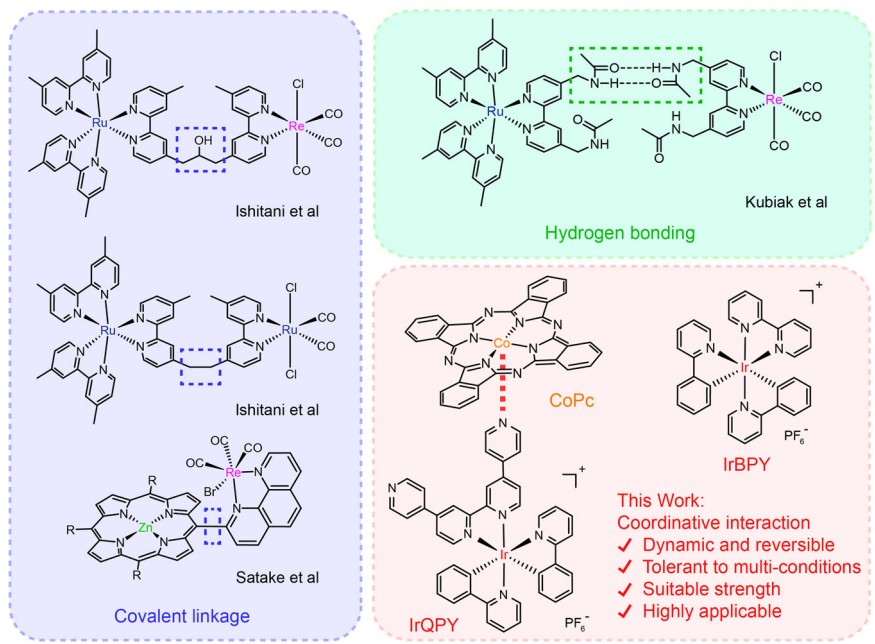

**Fig. 1 Comparison of different methods to bridge PS and catalyst for CO$_2$ reduction.** The blue region displays the examples of covalent linkages between PS and catalyst from Ishitani et al.[20,22] and Satake et al.[24]. The green region displays the example of hydrogen bonding between PS and catalyst from Kubiak et al.[35]. The red region introduces the coordinative interaction applied in this work, showing the used molecules and the merits of coordinative interaction.

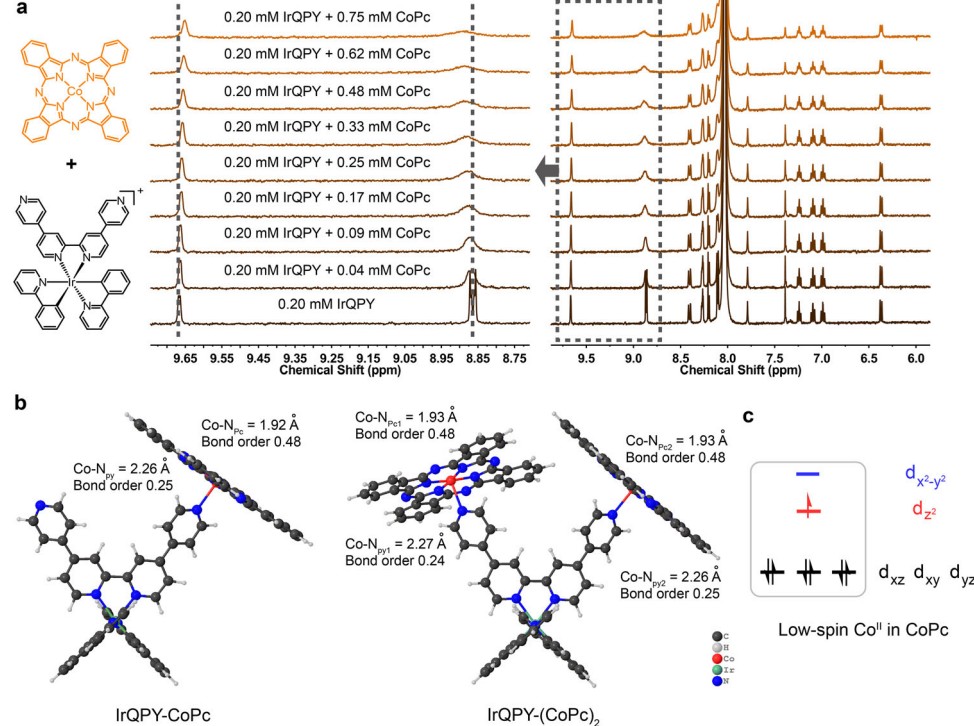

**Fig. 2 Characterization of coordinative interaction. a** $^1$H NMR titration with CoPc into a 0.20 mM DMF-d$_7$ solution of IrQPY, highlighting some shifted proton signals. **b** DFT-calculated structures of IrQPY-CoPc and IrQPY-(CoPc)$_2$. Counter anions are omitted for clarity. The Co-N bond lengths, bond orders, and **c** spin states of Co(II) are noted.

splitting, where the relatively differentiable shifts appeared at the pendant pyridyl protons of IrQPY. The proton shifts were then utilized in binding fits with different models and a 1:1 binding constant ($K_{11}$) of $2362 \pm 159\,\mathrm{M}^{-1}$ in the 2:1 non-cooperative model was calculated (see Experimental details and Supplementary Tables 2–4)[50]. The binding constant is significantly higher

than the Re-Ru H-bond-interacted system (ca. $300\,\mathrm{M}^{-1}$)[35], presumably thanks to the covalent nature of coordinative interaction. These observations suggest that IrQPY interacts with the paramagnetic CoPc via a dynamic interaction with the two pendent pyridyl moieties[31,35]. In contrast, the $^1$H NMR spectra of IrBPY did not change when titrated with CoPc (Supplementary Fig. 4),

which demonstrates negligible interaction between IrBPY and CoPc and indicates that the pyridine pendants in IrQPY are indispensable to interact with CoPc. Moreover, to preclude the interference of paramagnetic additive that may also cause the shifts without coordination, we further utilized the diamagnetic zinc phthalocyanine, ZnPc, to replace CoPc in titration. Similar shifted signals were observed in the IrQPY/ZnPc titration, with no peak broadening and splitting (Supplementary Fig. 5), which gives a $K_{11}$ of $1064 \pm 33\,\mathrm{M}^{-1}$ in the same 2:1 non-cooperative model (Supplementary Tables 5–7), at the same $10^3$ order as that of IrQPY/CoPc. This parallel outcome can partially exclude the magnetic interference in estimating the binding constant of the IrQPY/CoPc system. Therefore, the above $^1\mathrm{H}$ NMR titration results demonstrate the coordinative interaction between metal phthalocyanines and IrQPY via pendant pyridine groups.

**DFT calculation**. We further employed DFT calculation to mimic the coordinative interaction between IrQPY and CoPc at the M06[51]/BSI level (see Supplementary Information for details). The DFT-simulated structures of IrQPY, CoPc, and their two combinations, IrQPY-CoPc and IrQPY-(CoPc)$_2$, were taken into consideration. The calculation results are shown in the Appendix in the Supplementary Information. As indicated by Fig. 2b, the Co-N$_{py}$ bond lengths are ~2.26–2.29 Å in IrQPY-CoPc and IrQPY-(CoPc)$_2$, respectively, which are longer than those of equatorial Co-N$_{Pc}$ bonds (ca. 1.92–1.93 Å), showing the presence of weak axial coordination in both adducts. We also analyzed the Mayer bond orders[52] of Co-N bonds and found that the bond order of Co-N$_{py}$ (0.24~0.25) is only ca. half of that of Co-N$_{Pc}$ (0.48), which further indicates the weak coordinate bonds. Presumably, the low-spin $d^7$ Co$^{II}$ center in CoPc molecules owns a half-filled $d_{z2}$ orbital which will weaken the axial coordination from the $z$-direction, whereas the $d_{x2-y2}$ is vacant for strong equatorial coordination. In addition, the relatively labile axial coordinate bond can be partially attributed to the cationic nature of IrQPY, which makes the pyridine less electron-donating.

Besides geometry, IrQPY-CoPc and IrQPY-(CoPc)$_2$ adducts display calculated energy changes of $+1.9$ and $-0.8\,\mathrm{kcal\,mol}^{-1}$ relative to the free energies of isolated IrQPY and CoPc. Considering the computational error here is $\pm 2.0\,\mathrm{kcal\,mol}^{-1}$, the energy changes indicate the coordinative interaction between IrQPY and CoPc should be relatively reversible. As the free energy of IrQPY-(CoPc)$_2$ even slightly decreases with the coordination of one more CoPc, the coordination of two CoPc molecules is not markedly interfered by each other. Consequently, the energy results of coordinative interaction are in good accordance with the outcome of $^1\mathrm{H}$ NMR titration, which manifests the dynamic, reversible interaction between IrQPY and CoPc. At this stage, we have successfully evidenced the coordinative interaction between IrQPY and CoPc by corroborated experimental and theoretical results.

**Cyclic voltammetry**. Cyclic voltammetry was applied to investigate the redox properties of Ir PSs and CoPc and their interaction. Potentials are footnoted versus Fc$^+$/Fc (Fc = ferrocene) unless otherwise noted. Initially, the cyclic voltammogram (CV) of CoPc displays a quasi-reversible couple at $E_{1/2} = -1.28\,\mathrm{V}$ under N$_2$, assignable to a 1e Co$^{II/I}$ reduction (Supplementary Fig. 6). Upon the introduction of CO$_2$, a large catalytic wave appeared at ca. $-1.03\,\mathrm{V}$ with enhanced current initiated at an onset potential of ca. $-0.87\,\mathrm{V}$, which suggests catalytic CO$_2$ reduction. The addition of phenol (PhOH) led to a more positive onset potential (ca. $-0.68\,\mathrm{V}$; Supplementary Fig. 6) and a higher catalytic current, indicating that the presence of PhOH as a proton source can facilitate CO$_2$-reduction catalysis[53].

On the other hand, both CVs of IrQPY and IrBPY in CH$_3$CN exhibit three successive cathodic waves (Supplementary Fig. 7, 8). The first cathodic wave is attributed to the reduction of bpy-based ligands, and the latter two waves may originate from the reduction of ppy ligands[21,54]. Under an inert atmosphere, the addition of increasing [CoPc] in the solution of IrQPY cannot induce a notable shift for the reduction waves (Supplementary Fig. 7), showing that this electrochemical method is not effective to identify the weak interaction between two molecules. Overall, the redox properties of CoPc and Ir PSs were established for subsequent analysis of the photocatalytic mechanism.

**Photocatalytic CO$_2$ reduction with CoPc**. Visible-light-driven CO$_2$ reduction was carried out in a home-made quartz cell (Supplementary Fig. 9) with IrQPY or IrBPY as the PS (0.1 mM), CoPc as the catalyst (0.1 mM), PhOH (6.0 v%) as the optimized proton source, triethylamine (TEA; 2.5 v%), and 1,3-dimethyl-2-phenyl-2,3-dihydro-1H-benzo[d]imidazole (BIH; 80 mM) in dry CH$_3$CN, irradiated by $450 \pm 5\,\mathrm{nm}$ monochromic light. As shown in Fig. 3a, the IrQPY/CoPc system afforded a TON of $391 \pm 7$ and a 98% selectivity for CO production within 4 h, whereas the IrBPY/CoPc pair yielded a lower TON of $152 \pm 6$ and a decreased CO selectivity of 91%. More importantly, the maximum QE (see Fig. 3b and Methods) was calculated to $10.2\% \pm 0.5\%$ in the IrQPY/CoPc system within the first hour of photocatalysis, which is more than four times higher than that of IrBPY/CoPc ($2.4\% \pm 0.2\%$). The above comparison demonstrates the remarkable performance of the coordination-interacted IrQPY/CoPc system for photocatalytic CO$_2$ reduction.

To assure the contribution of coordinative interaction in the present system, the competitive binding from TEA and BIH should be considered. The $^1\mathrm{H}$ NMR titration between BIH and CoPc did not show a visible shift of the proton signals of BIH (Supplementary Fig. 10), which indicates the absence of coordination of BIH at CoPc. In contrast, $^1\mathrm{H}$ NMR titration is not available for TEA at low concentrations owing to the overlapped signals with DMF solvent background. However, the coordinative interaction between CoPc and IrQPY can be demonstrated by the blue shift from the UV–Vis spectra of CoPc before and after the addition of 2.5 v% TEA (Supplementary Fig. 11)[40,55]. It is thus meaningful to examine the competitive binding between TEA and IrQPY in the coordination to CoPc. We carried out the $^1\mathrm{H}$ NMR titration by adding CoPc into the DMF-$d_7$ solution of IrQPY in the presence of 2.5 v% TEA (Supplementary Fig. 12). The fitting results indicate that the coordinative interaction between CoPc and IrQPY survives albeit with a smaller 1:1 binding constant of $517 \pm 10\,\mathrm{M}^{-1}$ in the 2:1 non-cooperative binding model (Supplementary Tables 8–10), which is still higher than that of the precedent H-bond-interacted system (ca. $300\,\mathrm{M}^{-1}$)[35]. That is, the coordinative interaction between IrQPY and CoPc still exists in the real photocatalytic system despite the competition with TEA, showing tolerance to the reaction conditions. The remained interaction should be able to achieve the ISET for overall accelerated electron transfer in photocatalysis.

Although the IrQPY/CoPc system produced no CO after 4 h of reaction, the addition of fresh BIH instead of PS or catalyst could reinitiate the CO production with retention of most activity (Fig. 3c), manifesting that the ceased CO formation is mainly attributed to the consumption of BIH rather than PS or catalyst. Additionally, the photocatalysis under $^{13}$CO$_2$ atmosphere combined with mass spectroscopy showed the bare production of $^{13}$CO ($m/z = 29$; Supplementary Fig. 13), indicating that the evolved CO originates from the reduction of CO$_2$ rather than the decomposition of organic compounds in the IrQPY/CoPc system.

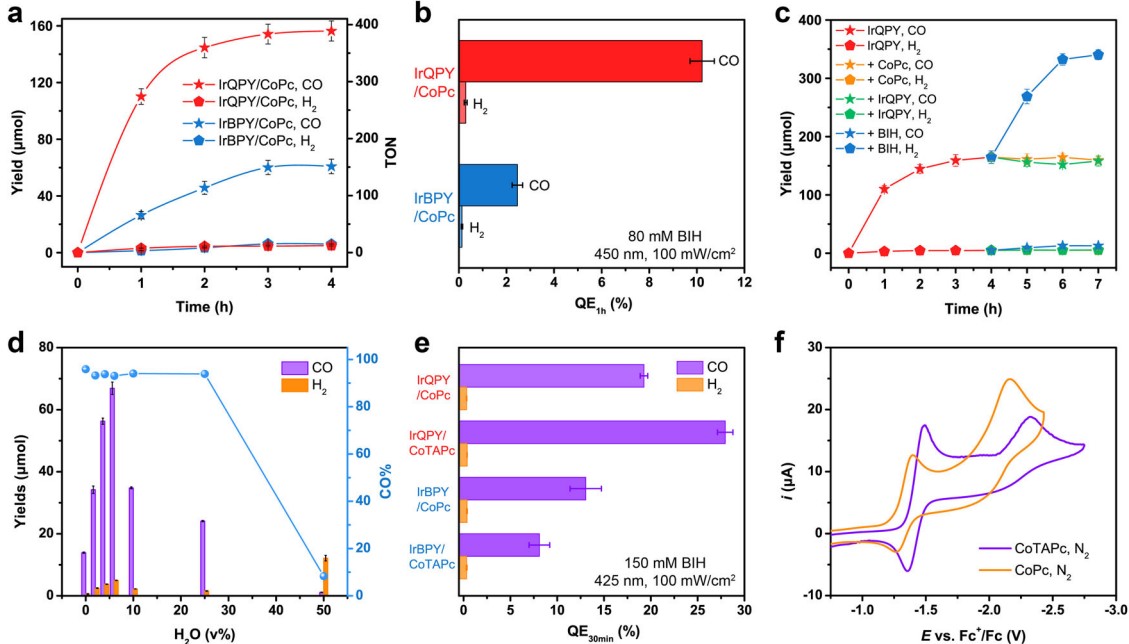

**Fig. 3 Photocatalytic CO₂ reduction. a** Time profiles of photocatalytic CO (star) and H₂ (pentagon) formation with IrQPY/CoPc (red) or IrBPY/CoPc (blue) system. **b** QEs for CO and H₂ formation from IrQPY/CoPc (red) and IrBPY/CoPc (blue) systems. **c** Time profiles of photocatalytic CO (star) and H₂ (pentagon) formation from IrQPY/CoPc (red) system. After 4 h, fresh CoPc (0.4 μmol, orange), IrQPY (0.4 μmol, green), or BIH (32 μmol, blue) was dispersed in 0.5 mL CO₂-saturated CH₃CN and injected into the solution, respectively. **d** Photocatalytic CO (violet) and H₂ (orange) yield with IrQPY/CoPc and varying volumes of H₂O, and corresponding CO% are shown (blue circle). **e** QEs for CO (violet) and H₂ (orange) formation from IrQPY/CoPc, IrBPY/CoPc IrQPY/CoTAPc, and IrBPY/CoTAPc systems with 150 mM BIH and 425 nm light within 30 min. **f** CVs of CoPc (orange) and CoTAPc (violet) under N₂. The error bars in the plots represent the standard deviations of three independent measurements.

The above results strongly confirm the robustness of IrQPY/CoPc system.

Each component is crucial to achieve high catalytic activity. Foremost, the absence of Ir PS, CoPc, CO₂, and light could not trigger the formation of CO (Supplementary Table 11, entry 1–4), whereas the use of low concentration CO₂, a simulated flue gas (10% CO₂/argon), is sufficient to initiate a substantial amount of CO, showing its potency to work under a low concentration of CO₂ (Supplementary Table 11, entry 5). Additionally, the addition of proton sources like PhOH, H₂O, or 2,2,2-trifluoroethanol (TFE), greatly improves the CO yields compared with the one under an anhydrous system (Fig. 3d, Supplementary Table 12 and Supplementary Fig. 14, 15), which is consistent with the CV of CoPc with PhOH (Supplementary Fig. 6). These enhancements by protons are attributable to the facilitated proton-coupled electron transfer during catalytic CO₂ reduction[53,56]. It should be noted that the CO selectivity of IrQPY/CoPc system can be impressively maintained at over 90% even upon the addition of water up to 25 v% volume (Fig. 3d and Supplementary Table 12), presumably endowed by the high selectivity of CoPc. However, further increasing the water ratio will lead to the severe precipitation of BIH and CoPc, leading to vanished activity. Thus, the use of a water-soluble electron donor and catalyst to develop this coordinatively interacted CO₂ reduction system is promising to attain high selectivity in a purely aqueous solution, which is a warranted work in the future.

$$\frac{I_0}{I} = 1 + K[Q] = 1 + k_q \tau_0 [Q] \qquad (1)$$

Further, two potential sacrificial electron donors, TEA and BIH, are also important to drive photocatalysis. As exhibited in Supplementary Table 12, In the absence of TEA, the TON for CO production decreased significantly to $167 \pm 2$ within 4 h. BIH is more indispensable as no CO formation took place without BIH. By measuring the fluorescent intensity and lifetimes (Supplementary Fig. 16) with a fluorescent spectrometer, the Stern-Volmer (S–V) plots between IrQPY and BIH or TEA can be obtained via Eq. 1[10], demonstrating a much higher apparent quenching constant ($k_q$) for BIH than that of TEA ($1.33 \times 10^{10}$ vs. $3.50 \times 10^8$ M⁻¹ s⁻¹; Supplementary Figs. 17 and 18), possibly due to the greater ease of BIH over TEA toward oxidation[6]. Rather than a sacrificial electron donor, TEA here should act as a base to deprotonate the oxidized BIH and thus achieve an irreversible electron transfer[57].

**Optimization for higher QEs.** We further tried to optimize the conditions to obtain higher QEs. We noticed that the significant consumption of BIH took place after 1 h of reaction, thus more BIH and shorter time were applied to obtained higher QEs (Supplementary Table 13). Next, the LED lights with different wavelengths, including 450, 425, and 405 nm, were also utilized for trials, showing that 425 nm is the optimal light source in our case, which gave a 19.2%±0.4% QE for IrQPY/CoPc system. The highest yield at 425 nm can be attributed to the relatively high-emission quantum yield (Φ) of IrQPY at this wavelength (Supplementary Table 14). More intriguingly, when a derivative of CoPc, cobalt(II) tetraamino-phthalocyanine (CoTAPc), was used instead under the above-optimized conditions, a high QE of 27.9%±0.8% could be attained under a solar-simulated light intensity of 100 mW cm⁻² at 425 nm (Fig. 3e). Remarkably, this QE value is record high among the molecular systems for photocatalytic CO₂-to-CO conversion employing noble-metal PSs and earth-abundant catalysts, surpassing Ru(bpy)₃²⁺/Fe(qpy')²⁺ (qpy', 2,2':6',2'':6'',2'''-quaterpyridine; QE = 8.8% at 450 nm)[58], Ru(bpy)₃²⁺/Ni(beptpy₂)²⁺ (QE = 11.1% at 450 nm)[17], Ru

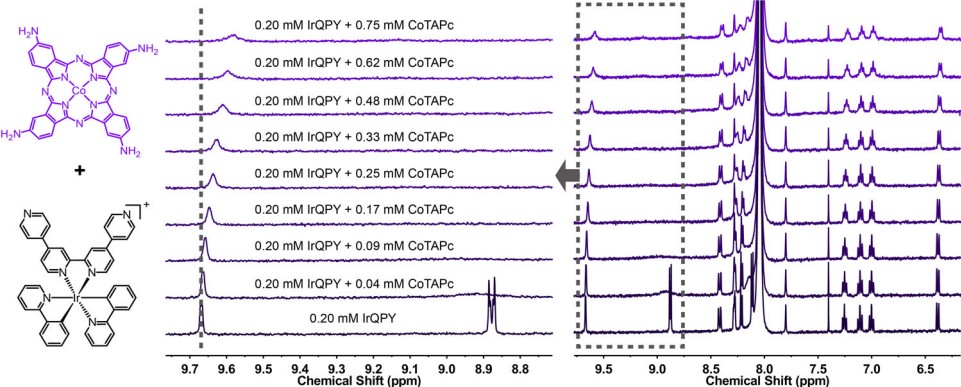

**Fig. 4 ¹H NMR titration.** ¹H NMR titration with CoTAPc into a 0.20 mM DMF-d₇ solution of IrQPY, highlighting some shifted pendent pyridyl proton signals.

(phen)₃²⁺/[CoZn(OH)L1]³⁺ (L1 = N[(CH₂)₂NHCH₂(*m*-C₆H₄) CH₂NH(CH₂)₂]₃N; QE = 4.9% at 450 nm)[12] and other pioneering systems listed in Supplementary Table 1.

To elucidate the origins of improvement, we first operated cyclic voltammetry to compare the redox properties of CoPc and CoTAPc. The comparison reveals that for CoTAPc, both its Co$^{II/I}$ reduction potential under N₂ (Fig. 3f) and its onset catalytic potential under CO₂ (Supplementary Fig. 19) are more negative than those of CoPc, attributable to the electron-donating amino groups. The more negative potentials indicate the higher overpotentials for catalysis, implying the disfavored photocatalysis[59]. The above electrochemical results are in agreement with the photocatalysis with IrBPY, where CoTAPc displayed a lower QE than CoPc under identical conditions (Fig. 3e). In sharp contrast, CoTAPc gave a higher performance than CoPc in the presence of IrQPY. This contradiction indicates that another factor remains to facilitate the photocatalysis with IrQPY rather than the altered intrinsic activity of catalyst. Then we conducted the ¹H NMR titration between IrQPY and CoTAPc to study if the intermolecular affinity is the more important factor (Fig. 4). Interestingly, the ¹H NMR spectra of IrQPY displayed more notable shifts upon the addition of CoTAPc instead of CoPc, and achieved a higher 1:1 binding constant of 3426 ± 220 M$^{-1}$ in the same 2:1 non-cooperative binding model (vs. 2362 ± 159 M$^{-1}$ from IrQPY/CoPc, Supplementary Tables 15–17), demonstrating a stronger coordinative interaction. The higher affinity can be explained that the coordinate bond is reinforced by sharing more electron density from the electron-donating amino groups. Based on the above results, we propose that the electron-rich amino groups of CoTAPc result in a higher overpotential for catalysis and a stronger binding with IrQPY. The latter feature can in turn overcome the negative influence of the former one, demonstrating that the coordinative interaction has a more important role in promoting photocatalysis over the capability of catalyst.

**Photocatalytic CO₂ reduction with other molecular catalysts.** More excitingly, this coordinative interaction strategy is widely applicable to other molecular catalysts with different metal centers and coordination modes. As a proof-of-concept, we then utilized a commercially available chloro iron(III) tetraphenylporphyrin (denoted as FePor; Fig. 5a) as another catalyst with a similar macrocyclic configuration and a varied metal center to CoPc, to cooperate with IrQPY in light-induced CO₂ reduction. As shown in Fig. 5b, under optimized conditions, the IrQPY/FePor system accomplished a TON of 253 ± 10, a

selectivity of 99.8%, and a QE of 1.7% for CO production. In contrast, the use of IrBPY only afforded a TON of 65 ± 5, one-fourth of the TON of IrQPY/FePor system under identical conditions. Also, the deactivation took place earlier than IrBPY/FePor system (4 vs. 7 h). The comparative results further demonstrate that the coordinative interaction strategy is versatile toward different molecular catalysts to promote their catalytic performances. Notably, the above performances of IrQPY/FePor system are superior to the reported photocatalytic systems with FePor, like g-C₃N₄/FePor (6 TONs, 99% selectivity)[60] and *fac*-Ir (ppy)₃/FePor systems (84 TONs, 79% selectivity)[61], displaying the boosted performance of FePor in cooperation with IrQPY.

Next, we envisioned that the use of a molecular catalyst with only one labile site, [Co(tpa)Cl]ClO₄ (CoTPA; tpa = tris(2-pyridylmethyl)amine; Fig. 5c)[56,62] as the example, maybe not applicable to the coordinative interaction method, as the only vacant site may possibly be seized by IrQPY to prevent CO₂ binding and catalysis. Unexpectedly, a similar enhancement in CO₂ reduction over H₂ evolution could also be observed in IrQPY/CoTPA in contrast to IrBPY/CoTPA combination (Fig. 5d), showing the successful application of coordinative interaction on the complex with only one accessible site. Meanwhile, the results also infer that during photocatalysis, the coordination is highly dynamic to avoid the stable occupancy at the labile sites, and merely serves as a bridge to accelerate the electron transfer from IrQPY to various types of catalysts, a typical ISET process, which is schematically proposed in Fig. 5e.

To verify the hypothesis, we added 0.1 M pyridine in the photocatalytic systems with IrQPY and IrBPY. Pyridine should be a stronger axial ligand relative to IrQPY due to its smaller size and neutral feature. Compared with the unperturbed IrQPY/CoPc system (Fig. 3a), the CO evolution rate and final yield were significantly inhibited by pyridine (Fig. 5f). In contrast, the pyridine-added IrBPY/CoPc system was not markedly affected, showing a similar CO yield of ca. 60 μmol to the one with no pyridine (Fig. 3a). Eventually, an even lower activity of IrQPY/CoPc than that of IrBPY/CoPc was observed in the presence of pyridine. That is, the pyridine coordination severely diminished the activity of IrQPY/CoPc, but not affected IrBPY/CoPc. This difference is most likely due to the pyridine-impeded coordination between IrQPY and CoPc, which further proves the positive contribution of the elastic coordination interaction between IrQPY and catalyst, and supports the hypothesis proposed in Fig. 5e. That is, the dynamic coordinative interaction with suitable affinity prevents the occupancy at the liable sites and barely serves a facile electron delivery to accelerate photocatalytic CO₂ reduction.

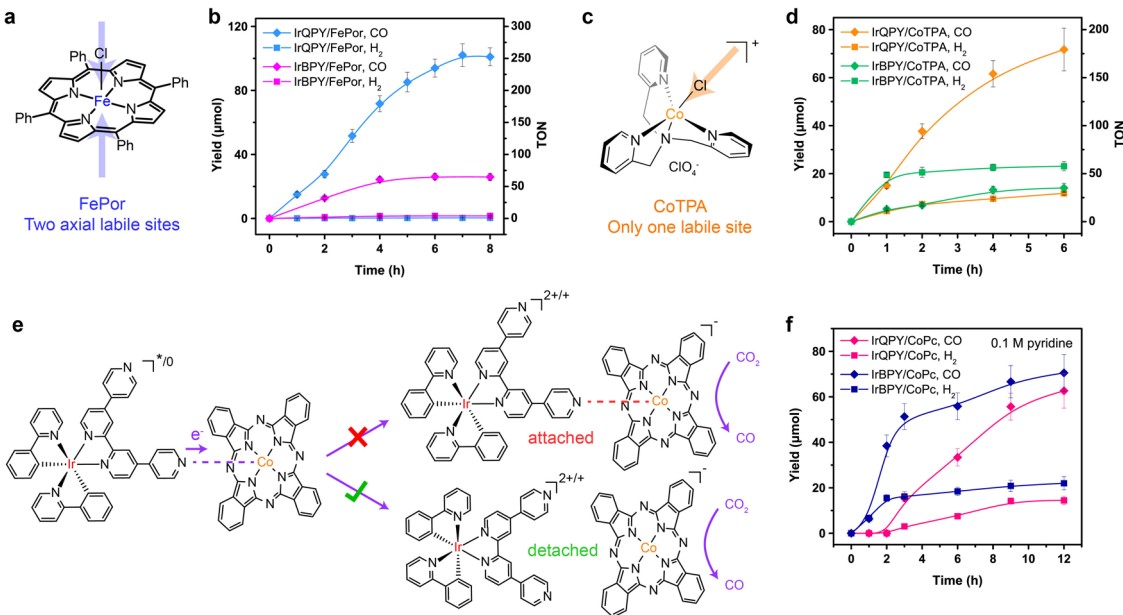

**Fig. 5 Wide applicability of dynamic coordination interaction. a** Structure of FePor. **b** Time profiles of photocatalytic CO (diamond) and $H_2$ (square) formation from a mixture of FePor (0.1 mM), IrQPY (blue, 0.1 mM) or IrBPY (magenta, 0.1 mM), TEA (2.5 v%) and BIH (80 mM). **c** Structure of CoTPA. **d** Time profiles of photocatalytic CO (diamond) and $H_2$ (square) formation from a mixture of CoTPA (0.1 mM), IrQPY (orange, 0.1 mM) or IrBPY (green, 0.1 mM), PhOH (5.0 v%), TEA (2.5 v%) and BIH (80 mM). **e** Proposed reaction mechanism for the relative states of IrQPY and CoPc during electron transfer and $CO_2$ reduction. **f** Time profiles of photocatalytic CO (diamond) and $H_2$ (square) formation from a mixture of pyridine (0.1 M), CoPc (0.1 mM), IrQPY (pink, 0.1 mM) or IrBPY (navy, 0.1 mM), TEA (2.5 v%), PhOH (6.0 v%) and BIH (80 mM). The error bars in the plots represent the standard deviations of three independent measurements.

**Photo-induced electron transfer pathway**. Finally, the photo-induced electron transfer pathway by quenching experiments was studied to verify the role of coordinative interaction in inter-molecular electron transfer. Compared with the one with BIH, the S–V plot between IrQPY and CoPc reveals an even steeper slope, resulting in a significantly higher quenching constant of $3.33 \times 10^{10} \, M^{-1} \, s^{-1}$ (Supplementary Fig. 20). This observation suggests that CoPc quenched the fluorescence of excited IrQPY more markedly than BIH. Thus the oxidative quenching pathway may be more accessible (Eqs. 2–4) than the reductive quenching pathway (Eqs. 2, 5, and 6)[63]. The same deduction can be made in the case of IrBPY according to the evaluation outcome of its quenching constants with BIH (Supplementary Fig. 21) or CoPc (Supplementary Fig. 22), respectively. However, the reductive quenching pathway may also be viable, as both the oxidative/reductive quenching constants are at the same order of magnitude ($\times 10^{10} \, M^{-1} \, s^{-1}$). And the light absorption by CoPc in quenching experiments should also be taken into consideration[64] according to the overlapped UV–Vis absorbance with IrQPY at 405–450 nm (Supplementary Fig. 23 and Supplementary Table 18).

Consequently, further verification was carried out by using time-resolved fluorescence spectroscopy under quenching conditions. As shown in Supplementary Figs. 24 and 25, the dynamic reductive quenching constants ($k_q'$) from time-resolved fluorescent quenching experiments are at the same magnitude as those in the steady-state measurements (listed in Table 2), while non-linear S–V plots were obtained for both Ir PSs with CoPc as the quencher. The poor linear behaviors and the small slopes (especially in the case of IrQPY) from the time-resolved fluorescent measurements suggest that the quenching behaviors of excited PSs by CoPc observed in the steady-state fluorescent spectroscopic measurements are not purely dynamic[64]. In turn, the reductive quenching pathway should be the main electron

transfer pathway in photocatalysis.

$$IrQPY + h\nu \xrightarrow{\phi} IrQPY^* \tag{2}$$

$$IrQPY^* + BIH \xrightarrow{k_{q,red}} IrQPY^- + BIH^{\cdot +} \tag{3}$$

$$IrQPY^- + CoPc \xrightarrow{k_{red}} IrQPY + CoPc^- \tag{4}$$

$$IrQPY^* + CoPc \xrightarrow{k_{q,ox}} IrQPY^+ + CoPc^- \tag{5}$$

$$IrQPY^+ + BIH/TEA \xrightarrow{k_{ox}} IrQPY + BIH^{\cdot +}/TEA^{\cdot +} \tag{6}$$

$$BIH^{\cdot +} + TEA \longrightarrow BI\cdot + TEAH^+ \tag{7}$$

$$IrQPY + BI\cdot \longrightarrow IrQPY^- + BI^+ \tag{8}$$

$$\frac{\tau_1}{\tau} = 1 + k_r\tau_1[Q] \tag{9}$$

In this context, it is necessary to further identify the electron transfer pathway. And if the reductive quenching pathway is dominant, the electron transfer rate should be determined from the reaction between the reduced PS and CoPc (Eq. 4). To this end, we implemented transient absorption (TA) spectroscopy measurements. As shown in Fig. 6a, upon excitation with 355 nm laser, an excited state absorption between 350 and 580 nm and strong bleaching between 580 and 750 nm were observed for IrQPY and the bleaching band well-matched with its emission spectrum (see the spectrum included in Supplementary Fig. 17). Moreover, its lifetime was determined to be 186 ns, close to the fluorescent emission of IrQPY (152 ns, Table 1), indicating the formation of the triplet excited state of IrQPY (IrQPY*). Next,

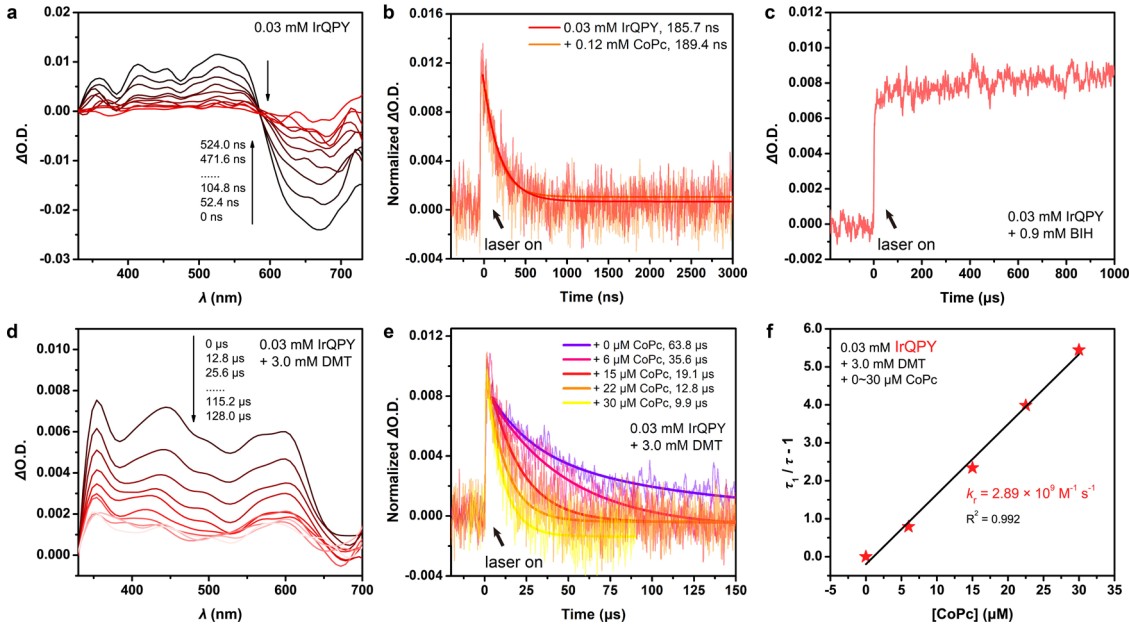

**Fig. 6 Transient absorption spectroscopy. a** TA spectrum of 0.03 mM IrQPY. **b** TA decay traces of 0.03 mM IrQPY in the absence (red) or presence (orange) of 0.12 mM CoPc at 530 nm. **c** TA decay trace of 0.03 mM IrQPY in the presence of 0.9 mM BIH at 350 nm. **d** TA spectrum of 0.03 mM IrQPY in the presence of 3.0 mM DMT. **e** TA decay traces of 0.03 mM IrQPY with 3.0 mM DMT in the absence (violet) or presence of 6 (pink), 15 (red), 22 (orange), 30 (yellow) μM CoPc at 350 nm. **f** Plot of ($\tau_1/\tau - 1$) versus [CoPc] with linear fitting for the IrQPY system. The excitation wavelength is 355 nm.

**Table 1 Redox properties and photophysical properties of Ir PSs.**

| Complex | $E_{red,1}$ (V) | $E_{red,2}$ (V) | $E_{red,3}$ (V) | $E_{ox}$ (V) | $\varepsilon_{450}$ (L mol$^{-1}$ cm$^{-1}$) | $\Phi_{450}$ (%) | $\tau_0$ (ns) | Isosbestic point (nm) | $E_{0-0}$ (V) | $E_{q, red}$ (V) | $E_{q, ox}$ (V) |
|---------|------|------|------|------|------|------|------|------|------|------|------|
| IrQPY | −1.37 | −1.65 | −2.11 | 0.96 | 1500 | 6.86 | 152 | 510 | 2.43 | 1.06 | −1.47 |
| IrBPY | −1.34 | −1.78 | −2.46 | 0.93 | 700 | 11.89 | 193 | 470 | 2.64 | 1.30 | −1.72 |

the triplet lifetime of IrQPY did not show significant change after the addition of four equivalents of CoPc (Fig. 6b), preliminary excluding the possibility of oxidative quenching in IrQPY/CoPc system. In contrast, under parallel conditions, the presence of excess BIH rendered another TA spectrum, which is different from that of IrQPY* (Supplementary Fig. 26), in which a positive, strong, and multiplet absorption band appeared in the range of 320–670 nm, indicating the formation of a new species. Furthermore, this species was kept with no exponential decay behavior within the detection time scale (Supplementary Fig. 26 and Fig. 6c), suggesting the formation of the long-lived reduced state of IrQPY, IrQPY$^-$. The sustained TA spectrum, however, precludes the estimation of the real-life of IrQPY$^-$, which should be caused by the use of BIH as a two-electron reductant. As noted by Ishitani et al.[65], the increased and sustained absorption among $\lambda = 320$–670 nm (at 350 nm in Fig. 6c) should be owing to the reduction of the ground-state IrQPY by the deprotonated product of BIH•$^+$, namely the highly reducing BI• ($-2.06$ V vs Fc$^+$/Fc; Eqs. 7 and 8)[65]. With two successive formation pathways of IrQPY$^-$ (Eqs. 3 and 8), the concentration and corresponding absorption of IrQPY$^-$ will not be diminished in an exponential decay process.

To address this issue, we utilized dimethyltoluidine (DMT) as a one-electron reductant to replace BIH in the TA studies[54]. As expected, a similar TA spectrum of IrQPY$^-$ can be obtained with DMT instead of BIH (Fig. 6d), in which a broad positive absorption band within 320–670 nm, mainly peaking at 350, 450, and 600 nm, was also noticed. Meanwhile, an ideal exponential decay trace can be afforded under these conditions to give the

lifetime of IrQPY$^-$ as 63.8 μs (Fig. 6e). More importantly, the corresponding TA spectrum (Fig. 6d) is consistent with the shape of the UV–Vis differential spectra obtained in spectroelectro-chemical (SEC) measurements under electrolysis at $E_{red,1}$ of IrQPY, whereas it is different from the one under oxidative electrolysis (Supplementary Fig. 27). The comparison between TA and SEC results strongly proves the dominant reductive quenching pathway involved in photocatalysis. Next, upon the addition of CoPc in the solution of IrQPY and excess DMT, the decay of IrQPY$^-$ was significantly faster, demonstrating the electron transfer from IrQPY$^-$ to CoPc. With varying [CoPc], the second-order reaction constant, $k_r$, can be determined as $2.89 \times 10^9$ M$^{-1}$ s$^{-1}$ with Eq. 9 in Fig. 6f (see Experimental details). On the other hand, parallel TA measurements were operated on IrBPY/CoPc/DMT system (Supplementary Fig. 28). The corresponding $k_r$ was determined as $1.11 \times 10^9$ M$^{-1}$ s$^{-1}$, much smaller than that of IrQPY/CoPc/DMT system. With the above $k_r$ values, the electron transfer rate constants, $k_{ET}$, of IrQPY$^-$/CoPc and IrBPY$^-$/CoPc are estimated as $2.89 \times 10^5$ and $1.11 \times 10^5$ s$^{-1}$, respectively (see Experimental details). On the basis of the close values of reducing force ($E_{red,1} = -1.37$ vs. $-1.34$ V) and longevity ($\tau_1 = 63.8$ vs. 63.1 μs) between IrQPY$^-$ and IrBPY$^-$, this sharp contrast in $k_r$ clearly demonstrates the merit of coordinative interaction between IrQPY and CoPc in accelerating the electron transfer.

With the above spectroscopic results (the values are listed in Table 2), we now can analyze the origins of the high QE from the coordinatively interacted systems. Following the reductive quenching pathway, three parameters related to Ir PSs are

**Table 2 Related data and calculated reaction constants from steady-state/time-resolved fluorescent quenching and TA experiments in degassed $CH_3CN$.**

| PS | Steady-state fluorescent quenching | | | Time-resolved fluorescent quenching | | TA | | | |
|---|---|---|---|---|---|---|---|---|---|
| | $\tau_0$ (ns) | $k_{q,ox}$ ($\times 10^9\,M^{-1}\,s^{-1}$) | $k_{q,red}$ ($\times 10^9\,M^{-1}\,s^{-1}$) | $k_{q,ox}'$ ($\times 10^9\,M^{-1}\,s^{-1}$) | $k_{q,red}'$ ($\times 10^9\,M^{-1}\,s^{-1}$) | $\tau_0'$ (ns) | $\tau_1$ ($\mu s$) | $k_r$ ($\times 10^9\,M^{-1}\,s^{-1}$) | $k_{ET}$ ($\times 10^5\,s^{-1}$) |
| IrQPY | 152 | 33.3 | 13.3 | N.A. | 12.2 | 186 | 63.8 | 2.89 | 2.89 |
| IrBPY | 193 | 20.3 | 18.1 | N.A. | 13.4 | 258 | 63.1 | 1.11 | 1.11 |

responsible for the QE of photocatalysis, including the emission quantum yields of Ir PSs ($\Phi$, Eq. 2), the reductive quenching rates between Ir PSs and BIH ($k_{q,red}$), and the reaction rate constants between the reduced Ir PSs and CoPc ($k_r$). First, although the UV–Vis spectra show that the absorption at 450 nm of IrQPY is higher than that of IrBPY (1500 vs. 700 L mol$^{-1}$ cm$^{-1}$; Table 1 and Supplementary Fig. 29), the $\Phi_{450}$ of IrQPY is lower than that of IrBPY (6.86% vs. 11.89%, Table 1), which are attributable to the more conjugated qpy ligand[66]. Second, the $k_{q,red}$ values have been estimated by the S–V plots with BIH titration, and both the steady-state and time-resolved fluorescent quenching measurements demonstrate the smaller $k_{q,red}$ values for IrQPY than those of IrBPY, consistent with the more oxidative IrBPY* and IrQPY* ($E_{q,red} = 1.30$ vs. 1.06 V). Although the former two parameters of IrQPY are inferior to IrBPY, the third one, $k_r$, is much more superior (2.89 vs. $1.11 \times 10^9\,M^{-1}\,s^{-1}$), confirming the decisive role of intermolecular electron transfer rate between PS and catalyst in photocatalytic performance, which is contributed by the introduction of dynamic coordinative interaction. Overall, the above results validate that although IrQPY possesses a lower $\Phi$ and a less oxidative excited state in comparison with those of IrBPY, the coordinative interaction between IrQPY and CoPc enables a much faster intermolecular electron transfer and thus achieves considerable QEs for photocatalytic $CO_2$ reduction.

## Discussion

In conclusion, a coordinative interaction between PS and catalyst has been employed in a purely molecular system for photocatalytic $CO_2$ reduction. The redox properties, molecule dynamics, and photo-induced electron delivery were carefully studied to reveal the role of coordination interaction in the photocatalytic mechanism. The results manifest that the dynamic coordinative interaction, in our case, not only accelerates the intermolecular electron transfer but also avoids the stable occupancy at the labile sites, which impedes substrate binding. Eventually, the coordinative interaction between the optimized PS and catalyst have boosted the QE from 2.4% to 27.9% for selective $CO_2$-to-CO conversion, an over 11-times improvement. Our studies demonstrate that in contrast to the documented covalent strategies, the application of coordinate bond has multiple advantages, including its dynamic feature, suitable affinity, extensive applicability, strong tolerance to conditional changes, etc. We believe that our work provides a versatile method to facilitate electron transfer and readily improve the photocatalytic efficiency of $CO_2$ reduction.

## Methods

**Materials.** IrBPY[66], BIH[67], and CoTPA[56] were synthesized according to the literature methods. Milli-Q ultrapure water (>18 MΩ) was utilized unless otherwise stated. CoPc (97%, β-form, Aldrich), FePor (97%, Shanghai Tensus Biotech Co., Ltd), CoTAPc (95%, Jilin Chinese Academy of Sciences-Yanshen Technology Co., Ltd.), PhOH (99.5%, Aladdin), TFE (99.5%, Aladdin), pyridine (99.5%, Aladdin), $CO_2$ (99.999%), $N_2$ (99.999%), $^{13}CO_2$ (99.9%), DMF-$d_7$ (99.5%, Acros) and other chemicals were commercially available and used without further purification.

**Instruments.** $^1$H NMR spectra were obtained on a Bruker advance III instrument (400 MHz). The electron spray ionization mass spectra of metal complexes were collected on an Ion Mobility-Q-TOF High-Resolution LC-MS (Waters Corporation, Synapt G2-Si). Single-crystal X-ray diffraction data were collected at 150 K on an Agilent Technologies Supernova system with Cu/$K\alpha$ ($\lambda = 1.54178$ Å) radiation. Electrochemical measurements were carried out using an electrochemical workstation (CHI 620E). The irradiation experiments were carried out with a blue LED light (Zolix, MLED4). Gas chromatographic analysis was conducted on an Agilent 7820 A gas chromatography equipped with a thermal conductivity detector and a TDX-01 packed column, where the oven temperature was held constant at 60°C, the inlet and detector temperature were set at 80°C and 200°C, respectively. The quantitative calibration of gaseous products utilized the commercially mixed gas (5% and 15% $H_2$/CO/$CH_4$ in argon, respectively). The isotopic labeling experiment was conducted under $^{13}CO_2$ atmosphere and the gas in the headspace was analyzed by a quantitative mass spectrometer attached Agilent 7890 A gas chromatography. The liquid phase of the reaction system was analyzed by ion chromatography (Metrohm, 930 Compact IC Flex, Supp 5 anion column, $Na_2CO_3$/$NaHCO_3$ aqueous eluent) to detect the presence of formate. UV–vis spectra were collected on a Shimadzu UV-3600 spectrophotometer. The quantum yields of Ir PSs were carried out on a fluorescence spectrometer (F-7000, Hitachi, Japan) equipped with an integrating sphere, which was also reproduced on another fluorescence spectrometer (FLS 1000, Edinburgh Instruments LTD.). The quenching experiments were conducted on a modular fluorescent life and steady-state fluorescence spectrometer (FLS 980 or FLS 1000, Edinburgh Instruments LTD.). TA spectra were measured on the laser flash photolysis instrument (LP980, Edinburgh Instruments LTD.). All experiments were operated at room temperature (24~25°C).

## Data availability

Experimental details, supplementary figures, and data are available from the authors. Most data generated in this study are provided in the Supplementary Information/Source Data file. The data with CCDC number, IrQPY (2043015), contain the supplementary crystallographic information for this paper. The data can be obtained free of charge from the Cambridge Crystallographic Data Centre via http://www.ccdc.cam.ac.uk/. Source data are provided with this paper.

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

## Acknowledgements

We are grateful for the expertize of NMR titration from Dr. Kai Wu from Faculty of Chemistry and Chemical Biology, TU Dortmund University, the TA spectroscopic measurements and analysis by Ping Wang, Dr. Song Guo, and professor Zhi-Ming Zhang, and the fluorescence analysis supported by Ms. Yu-Xin Chen and the X-ray diffraction measurements supported by Dr. Xiao-Long Feng from the Instrumental Analysis and Research Center in Sun Yat-sen University. This work was financially supported by National Natural Science Foundation of China (21737006, 22076222, and 22036003), Guangdong Basic and Applied Basic Research Foundation (2020A1515110017 and 2021A1515012033), China Postdoctoral Science Foundation (2020M683020 and 2021T140759), Guangdong Provincial Key R&D Programme (2020B1111350002) and the Fundamental Research Funds for the Central Universities (20lgpy87).

## Author contributions

J.W.W. and G.O. conceived and designed this project, J.W.W. and L.J. performed the experiments, H.H.H. carried out the DFT calculation, J.W.W. and O.G. analyzed the data, J.W.W., H.Z. and O.G. wrote and revised the article. All authors participated in drafting the paper and gave approval to the final version of the manuscript.

## Competing interests

The authors declare no competing financial interests.
