## [Peer Review File · Nature Communications]

REVIEWER COMMENTS

Reviewer #1 (Remarks to the Author):

The article by Gangfeng Ouyang and co-workers describes the photocatalytic carbon dioxide reduction to mainly CO, driven by noble metal Ir-based photosensitizers and Co-based catalysts. The system is based on intermolecular electron transfer from the photosensitizer to the catalyst, as all functional components are individual molecular entities.

The authors claim a coordination bond between the IrQPY photosensitizer and the CoPc catalyst, enacted between the peripheral 4-pyridine groups of IrQPY and the cobalt centre.

Furthermore, it is proposed that this interaction accelerates the electron transfer and consequently leads to higher CO₂ to CO conversion quantum yields. Also, nearly complete CO selectivity is observed. By chemical modification of the Co-phtalocyanine catalyst with amino groups, an enhancement of the quantum yield is reported.

The authors conclude that dynamic coordinative interactions take place, which not only accelerate the intermolecular electron transfer, but also avoid the stable occupancy at the labile sites, hence hampering back electron transfer.

The article is well written and a nice introduction into the subject is given. The proposed strategy towards a photocatalytic system, which allows efficient electron transfer but at the same time prevents electron back transfer is highly desirable and of great importance - not only for the field of light-driven CO₂ reduction.

The experimental data presented, such as NMR, electrochemical data, or DFT calculations suggest an interaction, which takes place between IrQPY and CoPc. And there is a clear improvement of the quantum efficiency when IrQPY is used instead of IrBPY.

However, the presented data does not convincingly support the conclusion of a dynamic coordination interaction between the photosensitizer and the catalyst. The main argument for the dynamic coordination interaction is based on the NMR titration experiments, which are in general a good means to study this type of interaction. In the contribution at hand, nine ¹H-NMR spectra were taken upon addition of the catalyst to the photosensitizer and the directly observed NMR shifts δ were fitted to a non-cooperative 1:2 binding model. Typically, the shift differences ($\delta - \delta_0$) are taken to obtain a binding isotherm rather than δ , as done in the present manuscript. Furthermore, no equations (mathematical and chemical) or some discussion on the chosen binding model are given. In my opinion, referencing a review article is not enough as this does not reflect the actual experimental conditions. A comparative presentation of the relevant statistical output (residuals, covariance, error of the obtained association constant) of the different binding models and a discussion on the selection and validity of the chosen model and alternative models should be presented. The shift differences as apparent from the presented NMR spectra are small for the IrQPY/CoPc and IrQPY/CoTAPc titration studies, pointing to a weak association between photosensitizer and catalyst. With the reported association constant in the order of 10² M⁻¹ (dissociation constant on the order of 10⁻² M⁻¹) and the photosensitizer concentration on the order of 10⁻³ M, the binding probability is likely at the lower range and the addition of more than three catalyst equivalents maybe necessary to obtain more robust data.

¹H-NMR data should be assigned to the proton positions to allow a better understanding of which signals have been chosen for the fit and of which signals are influenced by the addition of the catalyst. Additionally, Stern-Volmer quenching experiments were conducted, which are not only suited to address the question of oxidative vs. reductive electron transfer quenching, as discussed by the authors but also to address binding between photosensitizer and catalyst.

The presented graphs (I₀/I vs [Q]) all start at 0. This makes no sense, as at [Q]=0 the fluorescence of the unperturbed fluorophore should be observed (I₀/I=1). Furthermore, dynamic vs. static quenching is not addressed for the Q=catalyst case. If association between photosensitizer and catalyst occurs, a combination between static and dynamic quenching can be expected. The linear dependence of I₀/I vs. [Q] points to either static or dynamic quenching as the major pathway. Thus, additional fluorescence lifetime experiments in the presence of varying amounts of quencher are recommended to shed light on the mechanism. In the case of static quenching, the obtained Stern-

Volmer constant should be compared with the association constant obtained from the NMR titration experiments.

The [Cu(DPEphos)(pbcP)](PF₆) complex appears to be new and thus NMR data should be provided alongside with the already reported elemental analysis and mass spectrometry data

Apart from these considerations some I have some minor comments and questions:

The screen shots of the NMR-titration fits are hard to read. Export of the obtained data into a spread sheet and proper presentation are recommended.

Figure 4b shows a QE for CO of about 20% for IrQPY and about 5% for IrBPY, but the text mentions 10.2% and 2.4% respectively. Please double check.

How was the photocatalytically obtained CO and H₂ quantified - GC-TCD, or GC-MS? A few words on the calibration are recommended.

In the SI "Determination of QEs for CO production" an equation 7 is mentioned, which I can't find in the text or the SI. Please double check.

Typo in "Synthesis of IrQPY"/ "Synthesis of [Cu(DPEphos)(pbcP)](PF₆)" last line/second last line -> correct is calculated.

SuppFig1. The probability level of the thermal ellipsoids is missing.

SuppFig 10. Are these two peaks around m/z 45 and if so, how are they assigned?

Reviewer #2 (Remarks to the Author):

see attached file

Reviewer #3 (Remarks to the Author):

In the manuscript entitled "Rapid Electron Transfer via dynamic coordinative interaction boosts quantum efficiency for photocatalytic CO₂ reduction" the authors have described a novel strategy for increasing photocatalytic efficiency. The strategy involves a coordinate-bonding interaction between the photosensitizer and the catalyst which accelerates the rate of electron transfer upon photoexcitation. The hypothesis has been proven utilizing a pyridine-appended Iridium photosensitizer (IrQPY) and cobalt-phthalocyanine catalyst which gave a CO₂ to CO conversion efficiency of ca. 10.2 %. The pendant pyridine moieties occupy the axial-sites of cobalt-phthalocyanine. Further, the introduction of an electron donating amino-group on cobalt-phthalocyanine further increases the photocatalytic conversion efficiency to ca. 27.9 %. The absence of such high photocatalytic conversion efficiencies in control-systems essentially proves the importance of the dynamic coordinate interactions. The generality of this concept has been demonstrated by utilizing another macrocyclic catalyst viz. chloro iron (III) tetraphenyl porphyrin with IrQPY-photosensitizer and control-photosensitizer.

Over all the manuscript has been written-well, the arguments have been well-supported by experimental data and theoretical calculations. The manuscript may be considered for publication if only the following comments are addressed satisfactorily.

Comments

1. The authors should clarify the choice of the irridium-based photosensitizers instead of other well-known photosensitizers mostly Ruthenium-based photosensitizers.
2. The authors have argued efficient photo-induced electron transfer (PET) as the mechanistic reason behind the enhanced photocatalytic conversion efficiencies. However, there are two problems:
 - (a) Quenching constants do-not always represent the actual electron transfer rate constant. Quenching constants are diffusion limited parameters with dimensions M⁻¹s⁻¹ while electron transfer rate constant is of the dimension of s⁻¹. The actual electron transfer rate constant can be calculated

using transient absorption spectroscopy measurements.

(b) Further, PET is an excited state phenomenon which should be calculated via fluorescence lifetime quenching experiments using a TCSPC setup. The authors should not consider the quenching constants obtained from steady-state fluorescence quenching measurements while describing a PET phenomenon.

3. The authors claim a reduced rate of back-electron transfer due to the labile nature of the interaction further facilitates photocatalytic conversion. Although the argument is intuitively correct, there is no quantitative data to support this claim. The authors should either measure the rate of BET by spectroscopy or refrain from using such claims.

4. Further insights can be obtained from the ΔG_{PET} and ΔG_{BET} values [free-energy change associated with the forward and back electron transfer process]. The authors should calculate these parameters. For reference: J. Phys. Chem. C 2018, 122, 15819-15825.

5. The authors should refer to the recent review article on supramolecular photoredox catalysis (ACS Catal., 2021,11,710-733)

Editor:

Thank you again for submitting your manuscript "Rapid electron transfer via dynamic coordinative interaction boosts quantum efficiency for photocatalytic CO₂ reduction" to Nature Communications. We have now received reports from 3 reviewers and, after careful consideration, we have decided to invite a major revision of the manuscript.

As you will see from the reports copied below, the reviewers raise important concerns. We find that these concerns limit the strength of the study, and therefore we ask you to address them with additional work. Without substantial revisions, we will be unlikely to send the paper back to review. In particular, the referees express substantial concerns over the evidence for dynamic association and, as a consequence, the existence of appreciable concentrations of bound photosensitizer-catalyst complexes under reaction conditions, among other concerns.

Reply: Thank you so much for giving us opportunity to revise our work for further improvement. In the revised version, we have made every effort to acquire effective data and analysis to meet the requirements from the reviewers. More robust and correct data for the dynamic association were obtained, which indicate the coordinative interaction remains under reaction conditions.

More importantly, for the low concentration of bound photosensitizer-catalyst complexes, we think it not the active intermediate for catalysis. Because the active sites of catalyst will be blocked in the form of adduct. This conclusion has already been suggested by the use of a CoTPA catalyst (Figure 5d) with only one labile site, and the addition of pyridine in control experiments (Figure 5f). Beside the conventional outer-sphere electron transfer (OSET) by the random collision between IrQPY and CoPc, the intermolecular electron transfer can be additionally facilitated in a manner of inner-sphere electron transfer (ISET) via the coordinative interaction as either a stable adduct or a transition state, overcoming the diffusion limit. The above ISET theory was proposed by the Nobel laureate Henry Taube (*J. Am. Chem. Soc.* **75**, 4118-4119 (1953)). In turn in our case, the dynamic coordinative interaction between IrQPY and CoPc mostly provides the transition state rather than the stable adduct, which should also be effective in facilitating the electron transfer.

Overall, we believe that our point-to-point response can address most issues raised by the reviewers.

Reviewer 1:

The article by Gangfeng Ouyang and co-workers describes the photocatalytic carbon dioxide reduction to mainly CO, driven by noble metal Ir-based photosensitizers and Co-based catalysts. The system is based on intermolecular electron transfer from the photosensitizer to the catalyst, as all functional components are individual molecular entities.

The authors claim a coordination bond between the IrQPY photosensitizer and the CoPc catalyst, enacted between the peripheral 4-pyridine groups of IrQPy and the cobalt centre.

Furthermore, it is proposed that this interaction accelerates the electron transfer and consequently leads to higher CO₂ to CO conversion quantum yields. Also, nearly complete CO selectivity is observed. By chemical modification of the Co-phtalocyanine catalyst with amino groups, an enhancement of the quantum yield is reported.

The authors conclude that dynamic coordinative interactions take place, which not only accelerate the intermolecular electron transfer, but also avoid the stable occupancy at the labile sites, hence hampering back electron transfer.

The article is well written and a nice introduction into the subject is given. The proposed strategy towards a photocatalytic system, which allows efficient electron transfer but at the same time prevents electron

back transfer is highly desirable and of great importance - not only for the field of light-driven CO₂ reduction.

The experimental data presented, such as NMR, electrochemical data, or DFT calculations suggest an interaction, which takes place between IrQPY and CoPc. And there is a clear improvement of the quantum efficiency when IrQPY is used instead of IrBPY.

However, the presented data does not convincingly support the conclusion of a dynamic coordination interaction between the photosensitizer and the catalyst.

1. The main argument for the dynamic coordination interaction is based on the NMR titration experiments, which are in general a good means to study this type of interaction. In the contribution at hand, nine ¹H-NMR spectra were taken upon addition of the catalyst to the photosensitizer and the directly observed NMR shifts δ were fitted to a non-cooperative 1:2 binding model. Typically, the shift differences ($\delta - \delta_0$) are taken to obtain a binding isotherm rather than δ , as done in the present manuscript. Furthermore, no equations (mathematical and chemical) or some discussion on the chosen binding model are given. In my opinion, referencing a review article is not enough as this does not reflect the actual experimental conditions. A comparative presentation of the relevant statistical output (residuals, covariance, error of the obtained association constant) of the different binding models and a discussion on the selection and validity of the chosen model and alternative models should be presented.

Reply: Thank you for your constructive suggestions. We have revised the experimental section for NMR titration (Page S2), added discussion on the optimal choice among the alternative models for each titration couple (Supplementary Table 4, 7, 10 and 17), supplemented a comparative presentation of the relevant statistical output like residuals, covariance, error of the obtained association constant, etc. (Supplementary Table 2, 3, 5, 6, 8, 9, 15 and 17) in the revised Supplementary Information.

The shift differences as apparent from the presented NMR spectra are small for the IrQPY/CoPc and IrQPY/CoTAPc titration studies, pointing to a weak association between photosensitizer and catalyst. With the reported association constant in the order of 10² M⁻¹ (dissociation constant on the order of 10⁻² M⁻¹) and the photosensitizer concentration on the order of 10⁻³ M, the binding probability is likely at the lower range and the addition of more than three catalyst equivalents maybe necessary to obtain more robust data.

Reply: Thank you again for your kind suggestions. As suggested, we have operated the ¹H NMR titration for the couples with notable interactions in our work by using over 3 equivalents of catalyst versus IrQPY (Figure 2a and 4, Supplementary Figure 5 and 12). We also utilized the shifts of the proton signals from qpy ligand in the binding fits, as those protons are the main participators in the coordinative interaction. The new robust data are presented in the revised Supplementary Information (Supplementary Table 2-10, 15-17).

3. H-NMR data should be assigned to the proton positions to allow a better understanding of which signals have been chosen for the fit and of which signals are influenced by the addition of the catalyst.

Reply: As suggested, we have implemented and combined the results of the 1D and 2D ¹H-¹H COSY NMR spectra of IrQPY in DMF-*d*₇ for the assignment of proton positions, which are shown in Supplementary Figure 1 and 3 (Page S6-S7).

4. Additionally, Stern-Volmer quenching experiments were conducted, which are not only suited to

address the question of oxidative vs. reductive electron transfer quenching, as discussed by the authors but also to address binding between photosensitizer and catalyst.

The presented graphs (I_0/I vs $[Q]$) all start at 0. This makes no sense, as at $[Q]=0$ the fluorescence of the unperturbed fluorophore should be observed ($I_0/I=1$).

Reply: Thank you so much for your suggestion. We now revise the vertical coordinate unit as ' $I_0/I - 1$ '.

5. Furthermore, dynamic vs. static quenching is not addressed for the Q =catalyst case. If association between photosensitizer and catalyst occurs, a combination between static and dynamic quenching can be expected. The linear dependence of I_0/I vs. $[Q]$ points to either static or dynamic quenching as the major pathway. Thus, additional fluorescence lifetime experiments in the presence of varying amounts of quencher are recommended to shed light on the mechanism. In the case of static quenching, the obtained Stern-Volmer constant should be compared with the association constant obtained from the NMR titration experiments.

Reply: Thank you for your helpful suggestions. As suggested, we have operated transient fluorescent spectroscopic measurements. The results for both Ir PSs in the presence of quenchers have shown the linear and non-linear S-V plots for the addition of BIH and CoPc, respectively (Supplementary Figure 24 and 25, Page S19-S20). These observations indicate that the reductive quenching pathway is dominant in each of the Ir PS based system. The decreased PL intensity in the steady-state experiments and lifetimes in the time-resolved measurements by CoPc should be due to its light absorption, according to the overlapped absorption at around 450 nm between IrQPY and CoPc (Supplementary Figure 23). The above discussion has been added in the revised manuscript in Page 16.

In this context, the quenching results of lifetime measurements indicate the S-V constants between Ir PSs and CoPc are not reasonable to compare with the association constant from NMR titration experiment.

6. The $[Cu(DPEphos)(pbcp)](PF_6)$ complex appears to be new and thus NMR data should be provided alongside with the already reported elemental analysis and mass spectrometry data

Reply: Thank you for your kind suggestion. However, as required by Reviewer 2, we decide to delete the paragraph related to the new Cu complex as it seems to be redundant.

7. Apart from these considerations some I have some minor comments and questions: The screen shots of the NMR-titration fits are hard to read. Export of the obtained data into a spread sheet and proper presentation are recommended.

Reply: As recommended, we remove the screenshots with the replacement of tables to include the exported data (Supplementary Table 2, 3, 5, 6, 8, 9, 15 and 17) in the revised Supplementary Information.

8. Figure 4b shows a QE for CO of about 20% for IrQPY and about 5% for IrBPY, but the text mentions 10.2% and 2.4% respectively. Please double check.

Reply: We apologize for this mistake. The correct values are 10.2% and 2.4%, respectively. The mistakes in the new Figure 3b and 3e are fixed in the revised manuscript (Page 10).

9. How was the photocatalytically obtained CO and H₂ quantified - GC-TCD, or GC-MS? A few words on the calibration are recommended.

Reply: As suggested, we add a few calibration details for GC-TCD in the Methods section (Page 20). However, as we only carried out the semi-quantitative identification of isotope-labelled gas contents on the GC-MS, no calibration was operated before.

10. In the SI "Determination of QEs for CO production" an equation 7 is mentioned, which I can't find in the text or the SI. Please double check.

Reply: We apologize for this mistake and the equation is now listed as Equation S1 in the revised Supplementary Information (Page S4).

11. Typo in "Synthesis of IrQPY"/ "Synthesis of [Cu(DPEphos)(pbcp)](PF6)" last line/second last line -> correct is calculated.

Reply: We apologize for this typo and fix the spelling at "calculated" now in the revised Supplementary Information (Page S2).

12. SuppFig1. The probability level of the thermal ellipsoids is missing.

Reply: The possibility is 50%, which has been added in the caption of Supplementary Figure 1 (Page S6)..

13. SuppFig 10. Are these two peaks around m/z 45 and if so, how are they assigned?

Reply: This is due to the enlarged spectrum from GC/MS and the high abundance of $^{13}\text{CO}_2$. Now we present the original spectrum showing only one signal around m/z = 45 (Supplementary Figure 13 in Page S13).

Reviewer 2:

In this paper, Wang et al report a new set of compounds for photochemical CO₂ reduction with substantial increases in quantum yield compared to previous systems. The advances are based on use of a modified iridium complex as photosensitiser in which there are two outward pointing pendant pyridine groups. This is initially used in conjunction with cobalt phthalocyanine (CoPc) as catalyst together with triethylamine and BIH as base and sacrificial reductant, respectively. The solvent is MeCN and phenol is added as acid so creating a buffered system when taken together with TEA. Further improvement was achieved by deploying pendant amino groups on the cobalt phthalocyanine. The work is supplemented by more other changes in the photosensitiser and in the catalyst. The conversion of CO₂ to CO is verified by isotopic labelling. The authors postulate that their success is the result of coordination of the CoPc by the pendant pyridine groups on the iridium catalyst. Although they amply demonstrate the benefit of the pendant pyridine groups, the theory that they are the result of dynamic coordination has not been demonstrated satisfactorily in my view. The issue that causes most concern is that the measured equilibrium constant is so small that the proportion of coordination complex is exceedingly low at the concentrations used for photocatalysis. Moreover, the triethylamine and BIH will compete for coordination of CoPc and are present in much higher concentrations. I recommend major revision to consider the speciation under the reaction conditions and how this maps not the measured quantum yields.

1. Abstract – Please include information about TON

Reply: As suggested, we have included the information of TON in the abstract (Page 2).

2. Introduction last sentence.

There have been numerous systems of separate photosensitizer and photocatalyst components that are effective for CO₂ reduction, although they have no obvious propensity for association. Thus it is necessary to prove that the current examples of separate components work by coordinative interaction. Note that such interaction has been postulated previously for the role of the sacrificial reducing agent TEA as a dynamic ligand coordinating to the zinc porphyrin (Perutz et al JACS 2006)

Reply: Thank you for your kind suggestion. We have modified the end of Introduction for a more accurate discussion and include the example you suggested (Page 5, ref. 27). By addressing this issue, we also noticed our lack of important references from Prof. Robin Perutz, which are also cited in the revised manuscript (ref. 7, 24 and 28).

3. Determination of binding constant: This needs some more explanation. Please specify precisely which equilibrium constant is being calculated. The plot in fig S2 is unclear to me. None of the chemical shifts in the upper section change with added CoPc. What is happening? I would expect a fit assuming 1:1 as well as 1:2 binding, with the aim of determining both binding constants if possible.

Reply: Apology for our unclear presentation. As suggested by you and Reviewer 1, we have removed the screenshots, and presented the input and exported data for NMR binding fits. Especially, the fit results of the alternate binding models are displayed with discussion on the choices. The above data are shown in Supplementary Table 2-10, 15-17 in the Supplementary Information.

4. I would also expect that TEA and BIH compete for binding CoPc. Since both are present in far higher concentration than Ir(QPY) there is an important issue with regard to the theory that is proposed for the mechanism of these reactions.

Reply: (1) As suggested, we have investigated the potential binding between CoPc and BIH/TEA by the following NMR and UV-Vis measurements. On one hand, the ¹H NMR titration experiment showed no shift when CoPc was added into the solution of BIH, indicating the absence of interaction between CoPc and BIH (added as Supplementary Figure 10 in Page S11).

(2) On the other hand, the ¹H NMR titration experiment with low concentration of TEA is not available due to the overlapped signals between TEA and remnant DMF solvent peaks. But the UV-Vis spectra of a CH₃CN solution of CoPc in the absence and presence of TEA clearly shows the shifted absorbance (added as Supplementary Figure 11 in Page S12), which can be attributed to the coordination of TEA at the CoPc. At this stage, we further operated the ¹H NMR titration experiment for IrQPY/CoPc in the presence of 2.5 v% TEA (added as Supplementary Figure 12 and Supplementary Table 15-17). It can be seen that notable shifts can still be observed, although a small binding constant of 517 M⁻¹ was obtained. These observations prove that the coordinative interaction can be impaired due to the presence of TEA, while it still exists and contributes to the inner-sphere electron transfer for facile electron delivery from IrQPY to CoPc. The above discussion has been added in the revised manuscript (Page 9-10).

5. Electrochemistry. The effect of CO₂ on the CV of Ir(QPY) with CoPc is very striking. Considering the relatively small equilibrium constant, this part would benefit from varying the CoPc concentration with and without CO₂ and from varying the CO₂ partial pressure. Table 1 should list the effect of CO₂ on the potentials.

Reply: Thank you so much for your kind suggestions. As suggested, we re-operated the CVs of IrQPY upon addition of increasing [CoPc] strictly under N₂ (added as Supplementary Figure 7 in Page S9).

However, negligible shift of the reduction waves of IrQPY can be observed along with the increasing [CoPc]. We noticed that the leakage of trace amount of air when the CoPc was added can induce significant shifts which was observed previously. Consequently, we decide to rewrite the electrochemical section by not discussing the interaction behavior from the CVs. We are in great apology for our carelessness and thank you for pointing out this potential problem. We are thus not going to acquire other required CVs like those with varying [CoPc] with CO₂ and those with varying CO₂ partial pressure, as these experiments will not help the discussion on the dynamic coordinative interaction in our work.

7. UV absorption spectra. I could see nothing about the absorption spectra apart from fig S21. It is critical to understand the contribution of each of the components CoPc, Ir(QPY) and BIH to the absorption at the photolysis wavelengths under the experimental conditions. Please show the absorption spectra and the contributions of each component at 450, 425 and 405 nm.

Reply: As required, we supplement the absorption data for CoPc, IrQPY and BIH in Supplementary Figure 23 with the Supplementary Table 18 listing the absorption contributions at 450, 425 and 405 nm in the revised Supplementary Information (Page S27).

8. Photocatalysis. The design of the cell for photocatalysis should be specified in the SI including the pathlength.

Reply: As required, we have supplemented the photo of the used home-made quartz cell with the pathlength in Supplementary Figure 9 (Page S14).

9. In the legend to figure 4, the word “dispersed” is used. Does this imply that something did not dissolve? If so, what was insoluble?

Reply: CoPc and CoTAPc are not fully soluble in a concentrated mixture (5 mM), but it became homogeneous when the above mixture was sonicated and diluted in the photocatalytic system. The above discussion has been added in the Experimental details in the revised Supplementary Information (Page S3).

10. Copper complex. The final part about the copper complex should be deleted. It has no place here unless it is reported in full.

Reply: As suggested, we have deleted the related contents about the Cu complex.

11. Speciation. Although the argument for coordinative binding of Ir(QPY) appears superficially convincing, there are important issues concerning speciation. With a 1:2 equilibrium constant of ca. 300 the proportion of bound CoPc-Ir(QPY) at the concentrations used is very small indeed. (If $K = \frac{[\text{adduct}]}{[\text{Ir(QPY)}][\text{CoPc}]^2}$ and the initial concentrations are 0.1 mM, the adduct concentration is 3×10^{-10} M). Moreover, BIH and TEA may also bond competitively – see above. It is essential to calculate the speciation in order to understand the photo-induced electron transfer pathway.

Reply: (1) As answered above, we have elucidated the robust coordinative interaction between IrQPY and CoPc even in the presence of excess TEA as a potential ligand to CoPc. Additionally, BIH is not coordinative to CoPc.

(2) More importantly, for the speciation issue, we agree that the amount of the stable adduct is very few in our case even though we re-calculated the binding constant to the order of 10^3 M^{-1} . However, we believe that the stable adduct is not the active intermediate during catalysis, as the active sites in CoPc will be

blocked by stable IrQPY coordination. This conclusion has already been suggested by the use of a CoTPA catalyst (Figure 5d) with only one labile site, and the addition of pyridine in control experiments (Figure 5f). Beside the conventional outer-sphere electron transfer (OSET) by the random collision between IrQPY and CoPc, the intermolecular electron transfer can be additionally facilitated in a manner of inner-sphere electron transfer (ISET) via the coordinative interaction as either a stable adduct or a transition state (*J. Am. Chem. Soc.* **75**, 4118-4119 (1953); *Acc. Chem. Res.* **8**, 264-272 (1975)). That is, in our case, the dynamic coordinative interaction between IrQPY and CoPc mostly provides the transition state rather than the stable adduct, which should also be effective in facilitating the electron transfer.

(3) In addition, we have to mention that Kubiak et al have reported a H-bond-interacted system for CO₂ reduction and the binding constant is approximately 300 M⁻¹ (*J. Am. Chem. Soc.* **141**, 14961-14965 (2019)), a much smaller value than that of IrQPY/CoPc, which still presents the significant contribution of weak interactions in photocatalysis.

12. Photo-induced electron transfer pathway. The authors propose a series of equations for the photoinduced electron transfer pathway but none of them shows the interaction between Ir(QPY) and CoPc. I can see only one of these equations where the proposed dynamic interaction may have a role, namely eq 5 (there is a marginal possibility for eq 4). However, I am unable to see how the high quantum yields are compatible with very small proportion of Co(Pc)-Ir(QPy) adducts.

Reply: By operating the transient fluorescent quenching experiments and transient absorption spectroscopy, the reductive quenching pathway (equation 4) is verified as the main photocatalytic mechanism. The measured electron transfer rate from IrQPY^{*} to CoPc is significantly higher than the one from IrBPY^{*} to CoPc (2.89 vs. 1.11 × 10⁵ s⁻¹), demonstrating the merits of the dynamic coordinative interaction in facilitating electron transfer for catalysis.

As answered above, the QE will not be limited by the very small proportion of Co(Pc)-Ir(QPy) adducts which are not the active species, while the transition states from the dynamic coordinative interaction facilitates the electron transfer via ISET. Meanwhile, we need to emphasize that beside the coordinative interaction for boosting electron transfer, the use of potent PS and catalyst is of fundamental importance in achieving high QE values. The excellent photophysical properties of IrBPY and high catalytic activity of CoPc form the basis of the high QE of IrQPY/CoTAPc system after multiple optimizations like ligand modifications, light sources, additives, etc.

Reviewer 3:

In the manuscript entitled "Rapid Electron Transfer via dynamic coordinative interaction boosts quantum efficiency for photocatalytic CO₂ reduction" the authors have described a novel strategy for increasing photocatalytic efficiency. The strategy involves a coordinate-bonding interaction between the photosensitizer and the catalyst which accelerates the rate of electron transfer upon photoexcitation. The hypothesis has been proven utilizing a pyridine-appended Iridium photosensitizer (IrQPY) and cobalt-phthalocyanine catalyst which gave a CO₂ to CO conversion efficiency of ca. 10.2 %. The pendant pyridine moieties occupy the axial-sites of cobalt-phthalocyanine. Further, the introduction of an electron donating amino-group on cobalt-phthalocyanine further increases the photocatalytic conversion efficiency to ca. 27.9 %. The absence of such high photocatalytic conversion efficiencies in control-systems essentially proves the importance of the dynamic coordinate interactions. The generality of this concept has been demonstrated by utilizing another macrocyclic catalyst viz. chloro iron (III) tetraphenyl porphyrin with IrQPY-photosensitizer and control-photosensitizer.

Over all the manuscript has been written-well, the arguments have been well-supported by experimental data and theoretical calculations. The manuscript may be considered for publication if only the following comments are addressed satisfactorily.

Comments

1. The authors should clarify the choice of the irridium-based photosensitizers instead of other well-known photosensitizers mostly Ruthenium-based photosensitizers.

Reply: According to a comprehensive review by Ishitani et al (*J. Photochem. Photobiol., C* **25**, 106-137 (2015).), compared to the representative Ru-based PSs like $[\text{Ru}(\text{bpy})_3]^{2+}$ or $[\text{Ru}(\text{phen})_3]^{2+}$, the Ir-based PSs, typically IrBPY in our case or Ir(ppy)₃, possess stronger reducing forces at their reduced states and higher stability. These merits may endow the higher applicability of the above Ir PSs and their derivatives in photocatalytic CO₂ reduction than the Ru-based PSs. The above discussion has been added in the revised manuscript (Page 6).

2. The authors have argued efficient photo-induced electron transfer (PET) as the mechanistic reason behind the enhanced photocatalytic conversion efficiencies. However, there are two problems:

(a) Quenching constants do-not always represent the actual electron transfer rate constant. Quenching constants are diffusion limited parameters with dimensions M-1s-1 while electron transfer rate constant is of the dimension of s-1. The actual electron transfer rate constant can be calculated using transient absorption spectroscopy measurements.

Reply: Thank you so much for your constructive suggestions. As suggested, we have tried to carry out the transient absorption spectroscopy measurements (Figure 6, Supplementary Figure 26 and 28). The results ascertain that the reductive quenching pathway is the real electron transfer pathway. The electron transfer rate constants of IrQPY/CoPc and IrBPY/CoPc are estimated as 2.89×10^5 and $1.11 \times 10^5 \text{ s}^{-1}$, respectively, showing the contribution of coordinative interaction. The above results are shown in the revised manuscript (Page 17-19).

(b) Further, PET is an excited state phenomenon which should be calculated via fluorescence lifetime quenching experiments using a TCSPC setup. The authors should not consider the quenching constants obtained from steady-state fluorescence quenching measurements while describing a PET phenomenon.

Reply: As suggested, we have implemented the fluorescence lifetime quenching experiments using a TCSPC setup (Supplementary Figure 24 and 25, Page S19-S20), and calculated the quenching constants obtained from the above time-resolved measurements which are shown in Table 3. The related discussion is added in Page 16 in the revised manuscript.

3. The authors claim a reduced rate of back-electron transfer due to the labile nature of the interaction further facilitates photocatalytic conversion. Although the argument is intuitively correct, there is no quantitative data to support this claim. The authors should either measure the rate of BET by spectroscopy or refrain from using such claims.

Reply: Thank you for your suggestions. As suggested, we tried to follow the calculation methods mentioned by the reference below (*J. Phys. Chem. C* 2018, 122, 15819-15825), while we noticed that the k_{PET} and k_{BET} are values obtained from the photo-excitation process, appropriate for the oxidative

quenching pathway. However, the transient absorption spectroscopic results (Figure 6, Page 17-19) verify that the reductive quenching pathway is dominant. That is, the electron transfer between PS and catalyst takes place between the reduced PS and CoPc, involving no photo-induced process. Consequently, we cannot measure the rate of BET between PS and CoPc by the present measurements and would rather remove the claims of the reduced back-electron transfer.

4. Further insights can be obtained from the Δ GPET and Δ GBET values [free-energy change associated with the forward and back electron transfer process]. The authors should calculate these parameters. For reference: J. Phys. Chem. C 2018, 122, 15819-15825.

Reply: As mentioned above, due to the changed electron transfer pathway, we cannot calculate the parameters for PET and BET, and thus remove the claims of the reduced back-electron transfer in the revised manuscript.

5. The authors should refer to the recent review article on supramolecular photoredox catalysis (ACS Catal., 2021,11,710–733)

Reply: Thank you for your suggestion and your suggested paper is now cited as ref. 33 in the revised manuscript.

Reviewers' comments:

Reviewer #1 (Remarks to the Author):

The revision of Ouyang and coworkers has addressed all points, raised in the previous review report. However, the data of the main technique, used to support the conclusions in parts seems to be erroneous.

Also, transient absorption data has been added but not discussed in terms of the spectral and temporal behaviour. Although the argumentation evolves around lifetimes, no experimental details are given of how they were obtained.

The poor presentation of parts of the data and obvious errors make me doubt that this revision has been carried out with the necessary care. Under these circumstances I cannot recommend this manuscript for publication.

In the following my comments are summarized:

- The experimental description of the NMR titration experiment notes that the fitting was done without taking the shifts of the host-only situation into account. However, the host-only data is present in the tables. To my understanding, the fitting is based on the shift difference between the host-only data and the data at variable guest concentrations. What is the reason for the decision to not use the host-only data?

- What are the peak split values given in the SuppTables 3,6, 9, and 16?

- The SuppTables 2, 5, 8, and 15 all claim to list the protons of the qpy ligand. But the shifts in SuppTable 2 are very different from the rest. More importantly, the values listed in the SuppTables 5,8, and 15 do not fit to the NMR-assignment in SuppFigure 1.

- The assignment of the NMR data in SuppFigure 1 does not make sense as a couple of positions in the depicted molecule are not assigned in the spectrum. And in some cases, the given assignments do not fit to the signal integral.

- Additionally, the NMR chemical shift, given in the synthesis part, seems to agree with the spectrum in SuppFigure 1 and the COSY but not with the NMR chemical shifts in the titration spectra (Figure 2 and SuppFigure 4, 5, and 12).

- A minor point is that the F1 trace in the COSY seems to be the projection rather than the 1D spectrum.

- The authors commented in the rebuttal that "The decreased PL intensity in the steady-state experiments and lifetimes in the time-resolved measurements by CoPc should be due to its light absorption, according to the overlapped absorption at around 450 nm between IrQPY and CoPc (Supplementary Figure 23)."

I do not see how the overlapped absorption would impede the lifetime measurements, which is based on the IrQPY emission signal (provided that the enough signal is obtained, which seems to be the case). It seems that the change in lifetime upon CoPc addition is negligible.

- Figure 6b is unclear to me. The lifetimes, given in nanoseconds, do not seem to fit to the abscissa scale in microseconds. Why do the kinetics start at 1000 μ s?

- Figure 6c is unclear to me. What does it represent? Why is there a sudden signal increase at about 200 μ s?

-I'm missing a general discussion on the TA data and its temporal evolution.

The authors write that "The sustained TA spectrum should be caused by the use of BIH as a two-electron reductant,⁶⁴ which precludes the estimation on the lifetime of the reduced IrQPY, IrQPY⁻." Why does a two-electron reductant (BIH) preclude the observation of TA spectra and at least the discussion of the temporal evolution?

- I'm missing an experimental part for the TA experiments.

- How were the lifetimes, reported in TA section, determined?

Reviewer #2 (Remarks to the Author):

see attached

Reviewer #3 (Remarks to the Author):

In the revised Manuscript "Rapid electron transfer via dynamic coordinative interaction boosts quantum efficiency for photocatalytic CO₂ reduction" the authors have provided to the point answers to all the concerns raised by the reviewers and have supported their claims using additional experiments and significantly modified the manuscript accordingly. The revised manuscript can be accepted for publication after correcting a minor error.

TCSPC is a time-resolved fluorescence measurement technique. The authors should use "Time-Resolved Fluorescence Measurements" instead of "Transient Fluorescence" measurements.

Response to Referees

Reviewers' comments:

Reviewer #1 (Remarks to the Author):

The revision of Ouyang and coworkers has addressed all points, raised in the previous review report. However, the data of the main technique, used to support the conclusions in parts seems to be erroneous. Also, transient absorption data has been added but not discussed in terms of the spectral and temporal behaviour. Although the argumentation evolves around lifetimes, no experimental details are given of how they were obtained.

The poor presentation of parts of the data and obvious errors make me doubt that this revision has been carried out with the necessary care. Under these circumstances I cannot recommend this manuscript for publication.

Reply: We deeply understand our unprofessional presentation of the new data and careless mistakes make you feel very comfortable and annoyed. With great apology, we beg you for one last chance to further check on our response, which was carefully prepared under experts' supervision. No matter what final decision you will make, we are in full gratitude for your comments as they really alarm us to be serious to our work and prompt us to enrich our professional knowledge.

In the following my comments are summarized:

- The experimental description of the NMR titration experiment notes that the fitting was done without taking the shifts of the host-only situation into account. However, the host-only data is present in the tables. To my understanding, the fitting is based on the shift difference between the host-only data and the data at variable guest concentrations. What is the reason for the decision to not use the host-only data?

Reply: We apologize for making such misunderstanding. As you say, the fitting is based on the shift difference between the host-only data and the data at varying guest concentrations, which are applicable to our case. We did use the host-only data to calculate the shift differences for the fitting, but we did not directly use the host-only data into fitting as the first data point. Accordingly, we now revised the related description in the Experimental details in the revised Supplementary Information (Page S2).

- What are the peak split values given in the SuppTables 3,6, 9, and 16?

Reply: Apology for our unclear expression. The "peak split" values are the chemical shifts of the doublet/quartet proton signals of qpy ligand. We now fix the expression in the Supplementary Table 2, 3, 5, 6, 8, 9, 15 and 16.

- The SuppTables 2, 5, 8, and 15 all claim to list the protons of the qpy ligand. But the shifts in SuppTable 2 are very different from the rest. More importantly, the values listed in the SuppTables 5,8, and 15 do not fit to the NMR-assignment in SuppFigure 1.

- The assignment of the NMR data in SuppFigure 1 does not make sense as a couple of positions in the depicted molecule are not assigned in the spectrum. And in some cases, the given assignments do not fit

to the signal integral.

- Additionally, the NMR chemical shift, given in the synthesis part, seems to agree with the spectrum in SuppFigure 1 and the COSY but not with the NMR chemical shifts in the titration spectra (Figure 2 and SuppFigure 4, 5, and 12).

- A minor point is that the F1 trace in the COSY seems to be the projection rather than the 1D spectrum.

Reply: We sincerely apologize for the severe mistakes caused by our haste. Accordingly, we now revise the ^1H NMR results in the Synthesis part, as well as Supplementary Figure 1 and 3 for a correct and clear assignment. The projection in the Supplementary Figure 3 is now replaced as the higher-resolution 1D spectrum.

Moreover, we correct the data in Supplementary Table 2, 5, 8, 15 which are wrongly presented due to the mistaken use of Excel. The correction does not interfere the fitting results as the shift differences remain the same.

- The authors commented in the rebuttal that "The decreased PL intensity in the steady-state experiments and lifetimes in the time-resolved measurements by CoPc should be due to its light absorption, according to the overlapped absorption at around 450 nm between IrQPY and CoPc (Supplementary Figure 23)."

I do not see how the overlapped absorption would impede the lifetime measurements, which is based on the IrQPY emission signal (provided that the enough signal is obtained, which seems to be the case). It seems that the change in lifetime upon CoPc addition is negligible.

Reply: We apologize for our improper discussion on this issue. After consultation with an expert, we now discuss this issue as *"the reductive quenching constants from time-resolved fluorescent quenching experiments are similar to the ones in the steady-state measurements (Table 2), while non-linear S-V plots were obtained for both Ir PSs with CoPc as the quencher. The poor linear behaviors and the small slopes (especially in the case of IrQPY) from the time-resolved fluorescent measurements suggest that the quenching behaviors of PSs by CoPc observed in the steady-state fluorescent spectroscopic measurements are not purely dynamic."* (Page 16)

- Figure 6b is unclear to me. The lifetimes, given in nanoseconds, do not seem to fit to the abscissa scale in microseconds. Why do the kinetics start at $1000\mu\text{s}$?

Reply: We are very sorry for this typo of the unit, which should be ns rather than μs . We now modify this mistake in Figure 6b in the revised manuscript.

- Figure 6c is unclear to me. What does it represent? Why is there a sudden signal increase at about $200\mu\text{s}$?

-I'm missing a general discussion on the TA data and its temporal evolution.

The authors write that "The sustained TA spectrum should be caused by the use of BIH as a two-electron reductant,⁶⁴ which precludes the estimation on the lifetime of the reduced IrQPY, IrQPY $^-$." Why does a two-electron reductant (BIH) preclude the observation of TA spectra and at least the discussion of the temporal evolution?

- I'm missing an experimental part for the TA experiments.

- How were the lifetimes, reported in TA section, determined?

Reply: We apologize for being unable to describe the data of transient absorption spectroscopy with sufficient profession, which makes you difficult to understand the results. We now consult some experts

for a better discussion on these data and present the following answers with care.

For clear presentation, we now normalize the initiation time as 0 and redraw Figure 6, and then discuss the temporal evolution for this part. Furthermore, for clear explanation, we also analyzed the sustained trace as *“The sustained TA spectrum, however, precludes the estimation on the real lifetime of IrQPY, which should be caused by the use of BIH as a two-electron reductant, as noted by Ishitani et al (J. Am. Chem. Soc. 140, 17241-17254 (2018)). That is, in our case, the increased and sustained transient absorption among $\lambda = 320-670$ nm (at 350 nm in Figure 6c) should be induced by the reduction of the ground state of IrQPY by the deprotonated product of BIH^{•+}, namely the highly reducing BI⁻ (-2.06 V vs Fc⁺/Fc; Equation 7 and 8). With two successive formation pathways of IrQPY (Equation 3 and 8), the concentration and corresponding absorption of IrQPY will not be diminished in an exponential decay process.”*

To sum up, we now fulfil the discussion on the TA data and its temporal evolution in the revised manuscript (Page 18), as well as the experimental details of TA measurements and lifetime determination methods in the revised Supplementary Information (Page S5).

Reviewer #2 (Remarks to the Author):

The authors have responded very constructively to the referees' comments and their paper is much improved as a result. However, the responses have revealed two issues of concern that need to be addressed. p. 6 and Tables S4 and S7 I am very confused by the equilibrium constants. The text on p. 6 says that the data were fitted with various models and a 1:1 binding constant was calculated of 2362 M⁻¹. The main text does not specify which model gave this value.

Reply: We are sorry for making the confusing presentation. We now specify the models that give the corresponding constants in the main text of the revised manuscript (Page 6, 10 and 13).

Looking at Table S4 the 2:1 model gave a value of K₁₁ of 2362±7 M⁻¹. If this is the model, why isn't a second binding constant listed K₂₁?

Reply: We note that the non-cooperative mode means that each of the multiple binding processes does not interfere with each other, in which the second binding constant (K_{12} or K_{21}) will be equivalent to the first binding constant (K_{11}). This notion is now added in the Experimental details in the revised Supplementary Information (Page S3).

The table suggests the real error bar should be 10% in which case this should be quoted as $(2.4 \pm 0.2) \times 10^3$ M⁻¹.

Reply: We are thankful for the suggestion to note the error bar of the binding constant. We now provide the binding constants with error bars in the revised manuscript (Page 6, 10 and 13) and Supplementary Information (Supplementary Table 3, 6, 9 and 16) as you suggested.

Please also make sure that the host and guest are clearly defined.

Reply: Thank you for your suggestion. We now define the host/guest more clearly by noting their identities in the Supplementary Table 2, 5, 8 and 15.

Elsewhere in the paper (p.10, 13) please also use an appropriate number of significant figures and error bars.

Reply: As suggested, we provide the error bars in the Figure 3 and 5.

Fig S9 shows the cell for photocatalytic CO₂ reduction. This is cylindrical. We are also told that the irradiated area was 0.8 cm² and the diameter of the cell was 1.5 cm. I calculate that the beam diameter was ca. 1 cm. This means that the path length varied substantially across the cell. Please explain how it is possible to calculate an absolute quantum yield with this arrangement and indeed claim rather small error bars. Is it assumed that all the light is absorbed; if so has this been checked?

Reply: Thank you for pointing out this issue. We did assume all the light is absorbed by the cell although there should be some light loss by using this set-up. We previously added this assumption in the Experimental details for QE determination in the revised Supplementary Information as “*Under these conditions, the light entering the reaction solution was considered to be fully absorbed by PS, suggesting the evaluated QE is a lower limit*” (Page S4). And your suggestion inspires us to design cubic cells for a better measurement in the next project.

Fig S13 I don't understand the two peaks at about m/z 44.8 and 45.2 in the right-hand spectrum. What is going on?

Reply: We are sorry for this confusing presentation. We were meant to answer the previous question of Reviewer 1 by providing both figures. Now we remove the right-hand spectrum for a clear presentation of the isotope labelling results in Supplementary Figure 13 (Page S13).

Fig S23 is calibrated in units of absorption coefficient rather than absorbance. Is this correct for all three species?

Reply: Yes, all the three species are calibrated in units of absorption coefficient.

Reviewer #3 (Remarks to the Author):

In the revised Manuscript “Rapid electron transfer via dynamic coordinative interaction boosts quantum efficiency for photocatalytic CO₂ reduction” the authors have provided to the point answers to all the concerns raised by the reviewers and have supported their claims using additional experiments and significantly modified the manuscript accordingly. The revised manuscript can be accepted for publication after correcting a minor error.

TCSPC is a time-resolved fluorescence measurement technique. The authors should use “Time-Resolved Fluorescence Measurements” instead of “Transient Fluorescence” measurements.

Reply: Thank you so much for your positive comments on our efforts in revising the manuscript. As suggested, we now revised the term of “transient fluorescence measurements” as “time-resolved fluorescence measurements” in the revised manuscript and Supplementary Information.

REVIEWERS' COMMENTS

Reviewer #1 (Remarks to the Author):

The authors have addressed all issues raised in the review and significantly improved the discussion and presentation of the data.

However, one minor point remains: the authors now give the chemical shift of the doublet/quartet signals of the QPY ligand in the Supplementary Tables 2, 3, 5, 6, 8, 9, 15 and 16. But NMR signal assignment does at least not list quartets. I presume that doublets of doublets are meant. The fitting results are not effected.

This reviewer has no further formal objections.

Reviewer #2 (Remarks to the Author):

The authors have clarified the issues that I raised and I am now satisfied that the paper is suitable for publication.

Robin Perutz